# CIFD: Controlled Information Flow to Enhance Knowledge Distillation

**Yashas Malur Saidutta**    **Rakshith S Srinivasa**    **Jaejin Cho**    **Ching-Hua Lee**
**Chouchang Yang**    **Yilin Shen**    **Hongxia Jin**
Samsung Research America, Mountain View, CA
{ym.saidutta, r.srinivasa, jaejin.cho, chinghua.l}@samsung.com
{c.yang1, yilin.shen, hongxia.jin}@samsung.com

## Abstract

Knowledge Distillation is the mechanism by which the insights gained from a larger teacher model are transferred to a smaller student model. However, the transfer suffers when the teacher model is significantly larger than the student. To overcome this, prior works have proposed training intermediately sized models, Teacher Assistants (TAs) to help the transfer process. However, training TAs is expensive, as training these models is a knowledge transfer task in itself. Further, these TAs are larger than the student model and training them especially in large data settings can be computationally intensive. In this paper, we propose a novel framework called Controlled Information Flow for Knowledge Distillation (CIFD) consisting of two components. First, we propose a significantly smaller alternatives to TAs, the Rate-Distortion Module (RDM) which uses the teacher's penultimate layer embedding and a information rate-constrained bottleneck layer to replace the Teacher Assistant model. RDMs are smaller and easier to train than TAs, especially in large data regimes, since they operate on the teacher embeddings and do not need to relearn low level input feature extractors. Also, by varying the information rate across the bottleneck, RDMs can replace TAs of different sizes. Secondly, we propose the use of Information Bottleneck Module in the student model, which is crucial for regularization in the presence of a large number of RDMs. We show comprehensive state-of-the-art results of the proposed method over large datasets like Imagenet. Further, we show the significant improvement in distilling CLIP like models over a huge 12M image-text dataset. It outperforms CLIP specialized distillation methods across five zero-shot classification datasets and two zero-shot image-text retrieval datasets.

## 1 Introduction

A decade ago Hinton et al. proposed a mechanism called Knowledge Distillation (KD), where the insights from a larger model (the teacher) are transferred to a smaller model (the student) [1]. The teacher model, on account of larger modeling capacity, is capable of learning complex relationships in the data and is thus better at performing its target task. KD attempts to help the training of the student model by showing it the insights learned by the teacher. Specifically, Hinton et al. suggested the transfer of "dark knowledge" in the logits of a teacher classifier to a student classifier by minimizing the Kullback-Leibler (KL) divergence between a softened version of the predicted probability distribution of a teacher and a student model. However, dark knowledge is only one way of quantifying the insights learned by the teacher. In general, how to quantify the insights and how to transfer them to the student model remain open questions.

38th Conference on Neural Information Processing Systems (NeurIPS 2024).

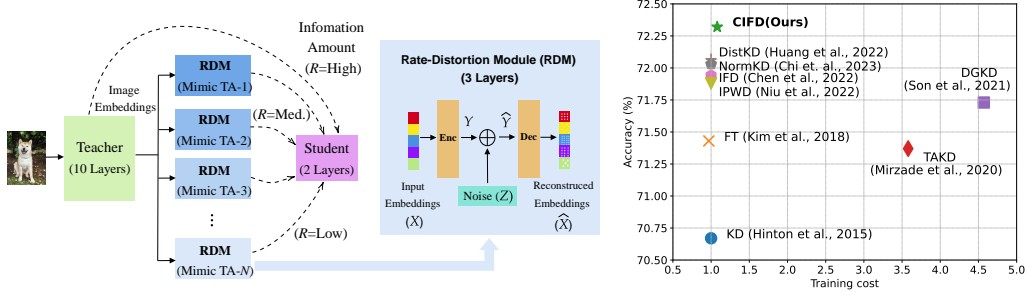

(a) Proposed CIFD for KD.    (b) CIFD vs. existing KD approaches.

Figure 1: (a) Proposed: Controlled Information Flow for Knowledge Distillation (CIFD). In CIFD, we explore the idea of Rate-Distortion Modules (RDM) that use the Teacher's embeddings to generate embeddings that act like Teacher Assistants (TAs) to distill knowledge to the Student. It does so by processing the embedding through a rate constrained communication channel. RDMs are much cheaper to train as they reuse the feature extractors learned by the teacher. By varying the rate constraint, RDMs can can simulate different TAs, and enable "TA" based training. (b) Training cost (normalized w.r.t. KD [1]) for distilling ResNet34 to ResNet18 over ImageNet. Earlier TA based approaches incurred a huge training cost increase due to the training of the TA models from scratch. Our proposed idea replaced TAs with RDMs and thus significantly reduces distillation cost while also improving performance.

Recently many researchers in the knowledge distillation community have been concerned with the gap between the teacher and student's modeling capacities [2, 3, 4, 5]. Going back to the human learning experience, we learn simple concepts first, intermediate concepts next, and finally advanced concepts. Based on this insight, prior works have proposed the use of Teacher Assistants (TAs), models whose size is between that of the teacher and the student, to help facilitate the knowledge transfer better [2, 3]. By distilling knowledge from the teacher to the assistant and then using the assistant to help the distillation into the student led to large performance improvement. However, as seen in Figure 1(b), these methods are around $3.5\times$ and $4.5\times$ more expensive than KD [1]. In this paper we introduce a method called Controlled Information Flow for Knowledge Distillation (CIFD) which is significantly cheaper to train. This brings the training cost of TA based methods closer to KD and other counterparts while obtaining state-of-the-art performance.

The proposed CIFD mechanism consists of two parts. First, we propose to process the teacher's penultimate layer embedding by a Rate-Distortion Module (RDM) to replace TAs. The RDM imposes a constraint on the amount of information through it. To replace multiple TAs of different capacities, we propose using RDMs with different levels of information constraints. Since RDMs operate on the teacher embeddings, they do not have to relearn low level features and can avoid the associated training and inference costs. As far as we know, this is the first application of Shannon's Rate-Distortion theory to aid knowledge distillation. However, as the number of RDMs increases, the student model tends to overfit to the teacher and assistants' outputs, as pointed out in [3] (also confirmed empirically here), thus reducing the generalization performance of the student model. Thus, to regularize the training of the student model in the presence of multiple RDMs, we propose the Information Bottleneck Module (IBM). The following are the contributions of this paper:

1. We propose the use of an RDM that takes the embeddings from a teacher model and mimics a teacher assistant. Since the RDM does not have to relearn low level features, it is two to three layers only; significantly smaller than a teacher assistant model.
2. We propose the use of the IBM in the student model during train-time. We find that IBM on its own provides benefits but is also a crucial regularizer as the number of RDMs increases.
3. On classification models, by distilling ResNet34 and ResNet50 (teachers) into ResNet18 and MobileNet-V2 (students), we achieve +1.66% (absolute) and +2.71% over KD [1], respectively.
4. To showcase the generality of the proposed CIFD, we apply it to distilling CLIP like models. Over three different teacher-student combinations, across five zero-shot classification and two zero-shot retrieval datasets, we find that our proposed method significantly outperforms CLIP specific distillation methods.

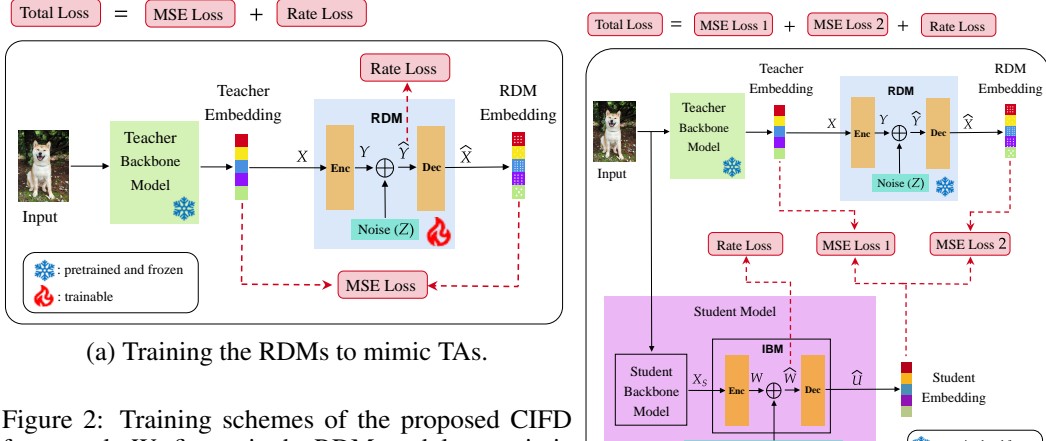

(a) Training the RDMs to mimic TAs.

Figure 2: Training schemes of the proposed CIFD framework. We first train the RDM modules to mimic teacher assistants as in (a). Then we train the student model using both the trained RDMs and the teacher model as in (b).

(b) Training the student model.

## 2 Related Works

**Knowledge Distillation:** Hinton et al. introduced the concept of transfering the knowledge from a large teacher model to a smaller student model using the logit space for classification models [1]. Many works have focused on different methods for transferring information like in the output space [6], transferring information in the intermediate layers of the teacher to the student [7, 8], to note a few. Some authors have focused on efficient distillation when only a subset of the original training data of the teacher is available [9], distilling easy classes first [10], or recently by looking at feature distillation as a diffusion process [11].

Recently, many works have pointed out the gap between the teacher and student models. Some authors looked at encoding the residual error between a large teacher and student [12, 13], using an ensemble of projectors for feature distillation [6], proposing a correlation based loss function in place of the KL divergence [9], and rectifying the imbalance at the concept level [4]. Most related to our work is the work of Mirzadeh et al. [2], and Son et al. [3]. Mirzadeh et al. proposed training teacher assistant models of progressively smaller size between that of a teacher and student model and using the smallest teacher assistant for distilling knowledge to the student. Son et al. argued that such a distillation process could cause error cascading and proposed using all the assistant(s) and the teacher for distillation. Further, each assistant is trained using all larger assistants and the teacher. This leads to progressively increasing computation costs as the number of assistants increases. Unlike their works, instead of explicitly training assistant models whose size is larger than the student, we propose a mechanism to mimic intermediate teaching assistants by training a small RDM module which is significantly smaller than even the student model. Further, unlike [3] where TAs need to be trained sequentially in the order of decreasing size, our RDMs are trained independently and in parallel. Another important work that is related to the RDM is the paraphraser network proposed by [14]. However we propose a more principled (from Shannon's insight on Rate-Distortion theory) loss function to train the RDMs where the information is compressed at different levels to mimic teacher assistants of different capacities.

The proposed IBM also has parallels in works like [6, 15]. The former looks at introducing an ensemble of projector networks that map the student embedding to the teacher embedding. Unlike the proposed IBM, which has a rate constrained loss function, the projector networks are trained for reconstruction error only. The concept behind IBM, the Information Bottleneck Principle (IBP), is also related to Masked-Image-Modeling [16] and Masked Generative Distillation [15]. The connections between IBP and improved generalization [17] support the success behind MIM and our proposed IBM. While our IBM loss function attempts to directly upper-bound each of the terms of the IBP objective, MIM upper bounds by dropping negative terms from the IBP objective. On the otherhand,

the MGD objective is an upperbound only under certain modeling assumptions. We discuss these connections in detail in Section 3.2 and Appendix C.

**CLIP Distillation:** Distillation of CLIP like models is a relatively new direction. One of the earliest works was to use CLIP to distill task specific knowledge into task specific student models [18]. In the task agnostic distillation of CLIP, TinyClip proposed a knowledge distillation and a pruning method for CLIP [19]. Yang et al. studied various losses for distillation including feature distillation and relational losses [20]. Relational losses attempt to maintain the same relative distances between embeddings in the teacher and student model [21, 22]. Another noteworthy paper is MobileCLIP which used an ensemble of teachers and a data refinement technique to obtain powerful small CLIP models [23]. However, the ideas proposed in these works are CLIP specific, i.e., they exploit the interactions between different modalities to compute the distillation losses. Unlike these works, our proposed idea is general, i.e., not CLIP specific.

**Rate Distortion and Information Bottleneck via Neural Networks:** Neural networks have been used to learn encoders and decoders for compression using Rate-Distortion theory [24, 25, 26] and communication over noisy channels [27]. On the other hand, Information Bottleneck has been used to improve neural networks in areas like improving generalization [28, 29], Out-of-Distribution detection [30, 31] and Out-of-Distribution generalization [32, 33].

## 3 Controlled Information Flow for KD

### 3.1 Controlling the information from the teacher using Rate-Distortion Theory

Shannon proposed Rate-Distortion Theory as a principled way to compress a signal. Given an input $X$, the goal of compression is to find a mapping from $X$ to its compressed version $\hat{X}$ such that $\hat{X}$ has minimal information about $X$ but at the same time the distortion does not exceed $D_0$. We can write this as an optimization problem of the form

$$\min_{X \to \hat{X}: D(X; \hat{X}) \leq D_0} I(X; \hat{X}), \tag{1}$$

where $D(\cdot, \cdot)$ is some distortion measure and $I(\cdot; \cdot)$ denotes the mutual information between two random variables [34]. We can convert this to an unconstrained optimization objective of the form

$$\min \quad R \cdot D(X; \hat{X}) + I(X; \hat{X}), \tag{2}$$

where $R$ determines the trade-off between information rate and distortion. A larger $R$ corresponds to more emphasis on minimizing the distortion at the cost of a higher information rate and vice-versa. This idea is best understood in the context of lossy compression where we wish to compress as much as possible while allowing tolerable distortion. However, rate-distortion theory is not only applicable to compression, but also to problems like Joint Source-Channel Coding where the compressed representation is subject to noise [34]. This case is represented in Figure 1(a). The encoded representation of the input is denoted as $Y$ and its noisy version as $\hat{Y}$. The independent noise added to encoded representation is denoted as $Z$.

However, even though Shannon's theory tells us what optimization problem to solve, it does not tell us how to solve it. When the encoders and decoders in Figure 1(a) are neural networks this problem is compounded because the objective cannot be computed ($I(X; \hat{X})$ is intractable). Instead, we can use variational approximations to compute an upper bound on the objective, which in turn will allow us to perform gradient descent to learn the encoder and the decoder. Let us denote $q(\hat{Y})$ as an approximation of true but unknown distribution of $\hat{Y}$, $p(\hat{Y})$. Then, we can compute an upper-bound on $I(X; \hat{X})$ as

$$I(X; \hat{X}) \leq I(Y; \hat{Y}) \leq H_q(\hat{Y}) + H(\hat{Y}|Y), \tag{3}$$

where $H(\cdot)$ denotes the entropy, and $H_q(\hat{Y})$ denotes the cross entropy computed using the distribution $q(\hat{Y})$. The first inequality follows from the Data Processing Inequality [34] and the second follows because cross-entropy is always greater than entropy [34]. Finally, note that $H(\hat{Y}|Y)$ is constant because the noise is independent of the encoder and decoder parameters.

The key here is to choose the approximating distribution $q$. Popular choices include the Gaussian distribution (e.g., Variational Autoencoders [35]), learning the distribution [36], or non-parametric approximations [24], we choose the latter. All these methods yield a mechanism where we can

compute $q(\hat{Y})$. Now, for simplicity let us assume that distortion measure $D$ is the $L_2$ norm, then we can learn the parameters of the encoder ($\Theta_e$) and the decoder ($\Theta_d$) by putting (3) into (2) as

$$\mathcal{L}_R = \mathbb{E}_{X,Z} \left[ R \left\| X - \hat{X} \right\|_2^2 - \log \left( q(\hat{Y}) \right) \right]. \tag{4}$$

Figure 1(a) shows how the RDMs are trained. Since the RDMs process extracted teacher embeddings, they do not have to relearn low level feature extractors from the raw input, thus making them significantly smaller than teacher assistants, which in turn makes them computationally cheaper.

In the case where we are distilling a classification model, an additional linear layer is used to convert the reconstructed embeddings to logits. Let $V$ represent the true classification label, $\hat{V}_T$ represent the teacher's output predictive distribution on the class labels, $\hat{V}_{RDM}$ be the same but as predicted by the RDM output. Then, the loss function used to train the RDM is

$$\mathcal{L}'_R = \mathbb{E}_{X,Z} \left[ \mathcal{L}_{CE}(V, \hat{V}_{RDM}) + \lambda_{KL} KL(\hat{V}_T || \hat{V}_{RDM}) \right] + \mathcal{L}_R. \tag{5}$$

Here, $\mathcal{L}_{CE}(\cdot, \cdot)$ is the cross-entropy loss, $KL(\cdot||\cdot)$ is the Kullback-Leibler divergence, and $\lambda_{KL}$ is weighting factor for the KL loss. For simplicity of writing we have dropped the presence of the temperature $\tau$ from the KL divergence term, however, it is assumed to be present. Figure 7 shows the illustration of how the RDM is trained for classification.

**Connections between Teacher Assistants and RDMs:** TAs limit the amount of information extracted from the input by using limited modeling capacity, i.e., smaller the model poorer the TA performance on the downstream task. On the other hand, we limit the amount of information extracted from the teacher embedding by passing it through a rate constrained communication channel. If we place a higher constraint on the information through the channel, our reconstructed embeddings will have more distortion compared to the teacher's and will perform poorly on the downstream task, just like the embeddings from a TA with a small model capacity. Thus, by choosing different values of $R$ in (4) we can mimic different TAs with different modeling capacities.

### 3.2 Controlling the information to the student using Information Bottleneck

In the previous section, we introduced RDMs to control the flow of information from the teacher model and mimic teacher assistants. However, when the teacher and multiple RDMs are providing feedback, the feedback can cause the student model to overfit and lead to poor performance. We can overcome this by constraining the information from the student model exposed to the feedback, i.e., provide partial feedback. Similar to the teacher model, we can accomplish this using another rate-constrained channel, but at the student. Further, this rate-constrained model is present only when there is feedback to the student, i.e., only during training. However, unlike the case of RDMs, we are not interested in trying to reconstruct the input to the rate constrained channel. Instead we wish to reconstruct the teacher or RDM embeddings. In Information Theory, the Rate-Distortion problem deals with compressing a random variable. However, when we want the compressed representation to be informative about another variable, it is called the Information Bottleneck problem. Thus, we call our proposed rate constrained module in the student model as Information Bottleneck Module (IBM).

Tishby et al. introduced the Information Bottleneck Principle (IBP) as a generalization to the Rate-Distortion problem [37]. Given an input $X_S$ and some random variable of interest $U$, the goal in IB is to find a representation $\hat{U}$ that removes as much information about input $X_S$ while retaining as much information about $U$. This is formulated as

$$\min \quad -I(U; \hat{U}) + \lambda_I I(X_S; \hat{U}), \tag{6}$$

where, $\lambda_I$ is the Lagrange multiplier.

Figure 1 shows the IB Module in the student model. $X_S$ represents the input, $W$ the encoded representation, $\hat{W}$ the noisy encoded representation, $Z_S$ the noise added during training only, and $\hat{U}$ represents the output of the IBM decoder. When working with neural network based encoders and decoders, both the mutual information terms in (6) are intractable. To overcome this setback, we use variational approximations, resulting in an upper bound on the IB objective (6) [28].

$$\mathcal{L}_I = \mathbb{E}_{X,Z_S} \left[ D(U, \hat{U}) - \lambda_I \log \left( r(\hat{W}) \right) \right], \tag{7}$$

where $D$ represents a suitable distortion metric, $r(\hat{W})$ is the approximation of the true distribution $p(\hat{W})$. In the IBM, $U$ is set to be the teacher's (or RDM's) embedding. So, at the output the IBM is attempting to reconstruct the teacher's (or RDM's) embedding. In case of a dimensionality mismatch, a projection layer is used. In practice, we do not use a dedicated encoder for IBM and let the student backbone model itself function as the encoder, i.e. $W = X_S$. The decoder is a simple network of at most one to two linear layers. The IBM is not trained separately, instead it is trained along with the student model.

**Connections to Masked-Image-Modeling:** In this discussion, we compare IBMs with Masked-Image-Modeling [16] and its extension into distillation Masked-Generate-Modeling [15]. In MIM, an input image ($T$) is masked ($T_M$). The masked image is fed through an encoder followed by an MIM encoder. The encoder provides an embedding of the image and the MIM encoder uses it to predict the tokens corresponding to the masked out part. The system is then trained to maximize the likelihood of the actual values of the masked out tokens. In Masked-Generative-Distillation, the masking is done after passing through the student backbone. Following that a generator attempts to predict the teacher's embedding (with a slight abuse of notation $U$). Let us denote the embedding from the student before the masking as $\hat{U}_{-M}$. Even though both MIM and MGD attempt to maximize the $\log p(U|\hat{U})$, we can easily show that this is equivalent to maximizing $I(U;\hat{U})$ (Lemma 1 in Appendix C). Following this we can write the following theorem to connect the MIM and MGD objectives to IBP.

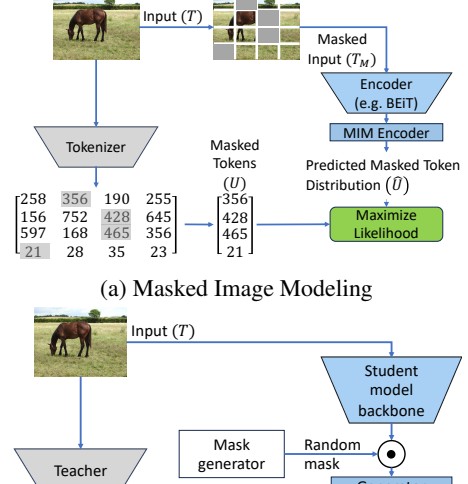

(a) Masked Image Modeling

(b) Masked Generative Distillation

Figure 4: Relation between Masked Image Modeling (MIM), Masked Generative Distillation (MGD), and Information Bottleneck Module (IBM) for Distillation.

**Theorem 1** (Informal). *The objective function of MIM is an upper bound on the objective function from IBP. The objective function of MGD is an upper bound on the objective function from IBP **if** $\hat{U}_{-M}$ is a discrete random variable and the mapping from $T$ to $\hat{U}_{-M}$ is deterministic.*

A more detailed explanation along with the formal statement and its proof are provided in Appendix C. The loss function of our IBM is directly written as an upper bound on the IBP objective. Unlike MIM or MGD, we do not drop the second term and instead use a non-parametric upper-bound to approximate it. This empirically should ensure a tighter approximation to the IBP objective than the other two. IBM works better because it forces the student to focus on those features necessary to predict the teacher embedding (first term) and remove information not useful in the prediction (second term), whereas in MIM and MGD the removal of information is either implicit or not present respectively. There have recently been works studying how IBP helps reduce generalization errors [17] which provides support that IBM in the student model should help reduce generalization errors. Our ablation studies in Section 4.3 also show similar results.

### 3.3 Controlled Information Flow for KD

With all the components in place, we can now derive the final loss function. Using the methodology described in Section 3.1, we assume that $N$ RDMs corresponding to different rate-constraints have been trained. To re-iterate, let $X$ be the input datapoint. Let us denote the teacher model's embedding as $X$; $\hat{X}_n$, $Z_n$ are the embedding and noise in the $n^{\text{th}}$ RDM respectively, and $\hat{U}$ corresponds to the student model's embedding. Also for sake of illustration, let us assume that the loss to match embeddings is the $L_2$ loss. Then, we can write the loss function for training the student model as

$$\mathcal{L}_{CIFD} = \mathbb{E}_{X, Z_1^N, Z_S}\left[ \left\|\hat{U} - X\right\|_2^2 + \sum_{n=1}^{n=N} \lambda_n \left\|\hat{U} - \hat{X}_n\right\|_2^2 - \lambda_I \log r(\hat{W}) \right], \qquad (8)$$

where $\lambda_n$ are weighting coefficients.

In the case of classification, let us denote the output predictive distribution of the teacher as $\hat{V}_T$, $\hat{V}_S$ for the student, $\hat{V}_n$ for the $n^{\text{th}}$ RDM, and the true label as $V$. Then, we can write the loss function for training the student model as

$$\mathcal{L}'_{CIFD} = \mathbb{E}_{X,Z_1^N,Z_S} \left[ \lambda_{CE}\mathcal{L}_{CE}(V,\hat{V}_S) + \lambda_{KL}KL(\hat{V}_T||\hat{V}_S) \right.$$
$$\left. + \lambda_{KL} \sum_{n=1}^{n=N} \lambda_n KL(\hat{V}_n||\hat{V}_S) \right] + \mathcal{L}_{CIFD}. \quad (9)$$

### 3.3.1. CIFD for CLIP style pretraining

CLIP (Contrastive Language-Image Pretraining) is a class of foundational models that are capable of embedding inputs from distinct modalities into a shared embedding space [38, 39]. In CLIP a modality specific encoder processes the input from a specific modality and embeds it into a shared embedding space. CLIP like models have shown tremendous performance in zero-shot classification, object-detection, and retrieval [38, 39]. Further, the trained encoders have also proved instrumental in powering Large Multimodal Models (LMMs) [40, 41, 42] and generative models [43, 44]. So distillation of these models has far reaching applications especially in on-device generative AI.

For simplicity, let us consider two modalities Image ($\mathbb{I}$) and Language ($\mathbb{L}$). Consider a batch of $B$ image-text pairs $\{(I^{(1)}, L^{(1)}), \ldots, (I^{(B)}, L^{(B)})\}$. Let $\hat{U}_{\mathbb{I}}^{(b)}$ denote the $L_2$-normalized embedding of the $b$-th image obtained from the image encoder and $\hat{U}_{\mathbb{L}}^{(b)}$ denote the same for the $b$-th text obtained from the language encoder (after IBM as in Figure 2(b)). Then we can write the contrastive loss from image to language embeddings as

$$\mathcal{L}_{CL,\mathbb{I}\to\mathbb{L}} = \frac{-1}{B} \sum_{b=1}^{B} \log \frac{\exp\left(\langle \hat{U}_{\mathbb{I}}^{(b)}, \hat{U}_{\mathbb{L}}^{(b)} \rangle / \tau\right)}{\sum_{k\in[B]} \exp\left(\langle \hat{U}_{\mathbb{I}}^{(b)}, \hat{U}_{\mathbb{L}}^{(k)} \rangle / \tau\right)}, \quad (10)$$

where $\langle \cdot, \cdot \rangle$ represents the inner-product between the two vectors. Using this we can write the loss to train CLIP-like models as

$$\mathcal{L}_{CL} = \mathcal{L}_{CL,\mathbb{I}\to\mathbb{L}} + \mathcal{L}_{CL,\mathbb{L}\to\mathbb{I}}. \quad (11)$$

The idea in contrastive loss is that embeddings of a paired image and text must be close to each other (high inner-product) when compared to embeddings of an image and/or text of non-pairs. Going forward we use CLIP to denote any CLIP like model.

Since the CLIP teacher has modality specific encoders, each encoder has its own set of RDMs. Each RDM is trained using (4). Let us denote the embedding for $b$-th image and text from the teacher by $X_{\mathbb{I}}^{(b)}$ and $X_{\mathbb{L}}^{(b)}$, respectively. We can define the $n$-th RDM embeddings as $X_{n,\mathbb{I}}^{(b)}$ and $X_{n,\mathbb{L}}^{(b)}$. Similarly, since the student has modality specific encoders, it has modality specific IBMs. Let us define $\hat{W}_{\mathbb{I}}$, $\hat{W}_{\mathbb{L}}$ (similar to $\hat{W}$ in Figure 1(a)) for the image and language encoder IBMs respectively. The output of the IBM decoder for the $b$-th image and text is denoted as $\hat{U}_{\mathbb{I}}^{(b)}$ and $\hat{U}_{\mathbb{L}}^{(b)}$, respectively. We can now write the modality specific CIFD loss for CLIP distillation as

$$\mathcal{L}_{CIFD,\mathbb{I}} = \frac{1}{B} \sum_{b=1}^{B} \left[ \left\| \hat{U}_{\mathbb{I}}^{(b)} - X_{\mathbb{I}}^{(b)} \right\|_2^2 + \sum_{n=1}^{n=N} \lambda_n \left\| \hat{U}_{\mathbb{I}}^{(b)} - X_{n,\mathbb{I}}^{(b)} \right\|_2^2 - \lambda_{I,\mathbb{I}} \log r_{\mathbb{I}}(\hat{W}_{\mathbb{I}}) \right], \quad (12)$$

Now we can write the final loss to perform distillation of CLIP using CIFD as

$$\mathcal{L}''_{CIFD} = \lambda_{CL}\mathcal{L}_{CL} + \mathcal{L}_{CIFD,\mathbb{I}} + \mathcal{L}_{CIFD,\mathbb{L}}, \quad (13)$$

where $\lambda_{CL}$ is weighting factor.

## 4  Experiments

**Experimental setup:** Our experimental results are split into two sections, one dealing with supervised classification on the CIFAR-100 [45] and Imagenet (IN) [46] datasets, and another with CLIP like

models trained on Conceptual Captions 12M dataset [47]. For the evaluation of the latter models, we conduct zero-shot classification on ImageNet, Imagenet-V2 (IN-V2) [48], Imagenet-A (IN-A) [49], Imagenet-R (IN-R) [50], and Object Net (ObjNet) [51], and we do zero-shot retrieval on COCO [52] and FlickR30k test set [53]. Architecture and training details in Appendix A. Unless otherwise mentioned results from other works are obtained from those papers.

## 4.1 Supervised Training

In Table 1 we first look at experiments on CIFAR-100 using a simple CNN model to specifically compare against the works of [2, 3] which only study these models. We see that using CIFD with just one RDM boosts performance over KD by +1.41%. However, when using 3 RDMs, we see a difference of +1.92% (absolute) over KD, and a boost of +0.63% over DGKD. DGKD uses three teaching assistants of size eight, six, and four layer CNNs. Since RDMs are less computationally intensive to train than teaching assistants, we trained two more and used them to perform distillation. We see that the resulting student model is +2.49% over KD, and +1.2% over DGKD. We present the more comprehensive results over CIFAR-100 in Table 2. We find that our proposed CIFD performs as good as existing methods.

However, the interest of this paper is more on large scale datasets. In Table 3 we look at the performance of our proposed CIFD methods in eq. (9) for distilling knowledge from a ResNet-34 to a ResNet-18 model and a ResNet-50 to a MobileNet-V1 model over the 1.28 million ImageNet dataset, respectively. We compare against multiple prior works listed in the table. We find that CIFD achieves State-of-The-Art (SoTA) performance, +1.66% (absolute) improvement over KD [1], +0.86% (absolute) over TAKD [2], and +0.5% (absolute) over DGKD [3] on ResNet-18. Looking at the case of ResNet-50 to MobileNet-V1, we find that the proposed method performs slightly above the work of [6] in top-1 accuracy, and achieves the SoTA performance in top-5 accuracy.

## 4.2 Image Language Pretraining

Finally, to establish the importance of the proposed method in the large dataset regime, we study knowledge distillation for foundational CLIP like models trained over the 12 million image-text pair dataset, CC-12M [47]. In Table 4, we look at the zero-shot classification and zero-shot retrieval results over multiple datasets for CLIP like models. We tested the distillation mechanism over different students and found that our distilled models beat the distillation mechanisms that are specialized to CLIP [19, 20]. To ensure a fair comparison, for TinyCLIP [19], we used only their knowledge distillation related innovation, not pruning (details in Appendix D.4). For ViT-B-16 models,

Table 1: Acc. (%) on CIFAR-100 over simple CNNs. All numbers from our implementation.

| Teacher | 10 layer CNN |
| Student | 2 layer CNN |
| --- | --- |
| Teacher | 54.42 |
| Student | 44.68 |
| KD [1] | 44.76 |
| TAKD (3 TA) [2] | 44.45 |
| DGKD (3 TA) [3] | 46.05 |
| CIFD (1 RDM) - Ours | 46.17 |
| CIFD (3 RDM) - Ours | **46.68** |
| CIFD (5 RDM) - Ours | **47.25** |

Table 2: Acc. (%) on CIFAR-100. Comparison with more baselines in Table 10.

| Teacher | WRN-40-2 | resnet32x4 |
| Student | WRN-40-1 | resnet8x4 |
| --- | --- | --- |
| Teacher | 75.61 | 79.42 |
| Student | 71.98 | 72.50 |
| AT [54] | 72.8 | 73.4 |
| FT [14] | 71.6 | 72.9 |
| KD [1] | 73.5 | 73.3 |
| CRD [55] | 74.1 | 75.5 |
| WSLD [56] | 73.7 | 74.8 |
| IPWD [4] | **74.6** | **76.0** |
| **CIFD (3 RDMs)** | **74.6** | **76.0** |

Table 3: Acc. (%) on ImageNet. Comparison with more baselines in Table 11.

| | Same arch. style | | Diff. arch. style | |
| --- | --- | --- | --- | --- |
| Teacher | ResNet-34 | | ResNet-50 | |
| Student | ResNet-18 | | MobileNet-v1 | |
| | **Top-1** | **Top-5** | **Top-1** | **Top-5** |
| Teacher | 73.31 | 91.42 | 76.16 | 92.87 |
| Student | 69.75 | 89.07 | 68.87 | 88.76 |
| KD [1] | 70.67 | 90.04 | 70.49 | 89.92 |
| FT [14] | 71.43 | 90.29 | — | — |
| TAKD [2] | 71.37 | 90.27 | — | — |
| DGKD [3] | 71.73 | 90.82 | — | — |
| IFD [6] | 71.94 | 90.68 | 73.16 | 91.24 |
| NormKD [57] | 72.03 | 90.64 | 72.79 | 91.08 |
| IPWD [4] | 71.88 | 90.50 | 72.65 | 91.08 |
| DistKD [5] | 72.07 | 90.42 | — | — |
| CIFD (3 RDMs) | **72.32** | **90.88** | **73.51** | **91.74** |

despite multiple attempts of [19], the training diverged. Looking at the results, in zero-shot classification, our distilled models show superior zero-shot classification over multiple datasets across three different teacher-student combinations. We see as high as 5.3% improvement over the nearest prior work over IN-R dataset on ViT-S-16 (Table 14). Similar trends hold over zero-shot image-to-text and text-to-image retrieval over both the COCO [52] (Table 15) and Flickr30k dataset [53], where our proposed models outperform existing distillation methods by as much as 8% (absolute) for I→T@5 on the COCO dataset on ViT-S-16. We see largest gains in performance when the ratio between teacher to student size is large.

Table 4: Zero-shot image-text classification performance on ImageNet and Object Net, and retrieval performance on FlickR30k [53] test sets. Zero-shot classification over more datasets and baselines in Table 14, zero-shot retrieval results over more datasets and baselines in Table 15. All results reproduced by us.

| Method | Model | IN(%) | | ObjNet(%) | | Flickr30k(%) | | | | | |
|---|---|---|---|---|---|---|---|---|---|---|---|
| | | | | | | Image → Text | | | Text → Image | | |
| | | Top-1 | Top-5 | Top-1 | Top-5 | R@1 | R@5 | R@10 | R@1 | R@5 | R@10 |
| Teacher | ViT-L-14 | 79.22 | 95.52 | 72.80 | 90.01 | 88.30 | 98.70 | 99.80 | 73.86 | 91.92 | 95.44 |
| Student | RN50 | 36.47 | 64.40 | 20.99 | 44.98 | 59.50 | 83.90 | 89.60 | 45.52 | 73.44 | 81.98 |
| OpenCLIP [58] | RN50 | 34.57 | 61.97 | 21.33 | 43.44 | 52.10 | 79.20 | 86.40 | 40.30 | 66.62 | 76.44 |
| TinyCLIP [19] | RN50 | 36.61 | 64.43 | 21.86 | 44.02 | 59.50 | 83.20 | 88.60 | 43.08 | 70.58 | 80.26 |
| CLIPKD [20] | RN50 | 46.32 | 75.77 | 27.54 | 51.66 | 61.90 | 85.80 | 91.10 | 49.92 | 77.80 | 85.62 |
| CIFD (1 RDM) | RN50 | **47.36** | **76.92** | **28.85** | **53.46** | **65.40** | **87.60** | **93.10** | **50.46** | **78.34** | **85.64** |
| TinyCLIP [19] | ViT-S-16 | 31.29 | 58.01 | 16.43 | 36.90 | 50.40 | 74.90 | 84.50 | 36.74 | 63.60 | 74.04 |
| CLIPKD [20] | ViT-S-16 | 39.42 | 69.71 | 22.11 | 45.45 | 54.70 | 81.20 | 88.30 | 46.26 | 73.96 | 82.68 |
| CIFD (1 RDM) | ViT-S-16 | **42.79** | **73.06** | **23.11** | **46.98** | **62.00** | **85.50** | **91.50** | **50.36** | **76.70** | **84.58** |
| CLIPKD [20] | ViT-B-16 | 51.25 | 79.91 | 29.63 | 53.44 | 69.50 | 89.50 | 93.90 | 55.94 | 82.42 | 88.56 |
| CIFD (1 RDM) | ViT-B-16 | **54.10** | **81.94** | **32.63** | **57.14** | **73.60** | **91.80** | **96.00** | **60.10** | **84.36** | **90.00** |

Table 5: Effect of number of RDMs on zero-shot image-text classification over ImageNet and Object Net, and retrieval over FlickR30k [53] test sets. Best in bold and second best underlined.

| Method | Model | IN(%) | ObjNet(%) | Flickr30k(%) | | | | | |
|---|---|---|---|---|---|---|---|---|---|
| | | Top-1 | Top-1 | Image → Text | | | Text → Image | | |
| | | | | R@1 | R@5 | R@10 | R@1 | R@5 | R@10 |
| CLIPKD [20] | ViT-S-16 | 46.78 | 27.23 | 66.30 | 86.60 | 91.60 | 50.92 | 77.14 | 85.42 |
| CIFD (1 RDM) | ViT-S-16 | _49.37_ | **29.49** | 66.90 | _88.80_ | _93.60_ | _53.32_ | 79.50 | 87.00 |
| CIFD (3 RDM) | ViT-S-16 | **49.98** | _29.37_ | **67.80** | **90.00** | **95.10** | **53.54** | **80.32** | **87.28** |

## 4.3 Ablation studies

**Do RDMs mimic Teacher Assistants?**

In Figure 5 we show the RDMs used to obtain the results in Table 1. As the information through the RDM is constrained, the RDM's output classification performance also falls. This is because as the information rate is constrained, and the RDM is forced to drop some features that are useful for classification. The corresponding TAs used by [2] in Table 1, have accuracies of $54.72\%$, $54.84\%$, $50.32\%$ for model sizes of eight, six, and four layer CNNs resepctively. This trend closely matches the RDM trend in Figure 5. Thus, by varying $R$ we can mimic teaching assistants of different modeling capacities, i.e., lower $R$ we mimic a smaller TA, higher $R$ we mimic a larger TA.

**How does number of RDMs affect performance? Does IBM help?** In Table 6, we study the effect of the number of RDMs and the effect of IBM in conjunction with the

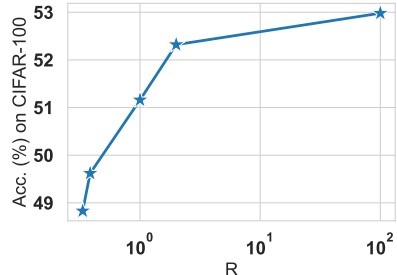

Figure 5: Effect of rate-constraint on RDM's classification performance. The graph behaves like a rate-distortion curve, information rate $R$ is proportional to performance.

RDMs. We find that increasing the number of RDMs without IBM initially improves the performance, but with five RDMs the performance on CIFAR100 drops. We hypothesized that this is due to overfitting, which DGKD proposed to overcome by KD dropout [3]. In KD dropout, the gradients from the teacher and teaching assistants are randomly cut off (Fig. 3 in [3]). Despite incorporating it, the performance without IBM does not show improvement. However, with IBM we see the expected benefits. Further, we see that IBM on its own also improves the performance of KD [1]. This further corroborated on Imagenet as shown in Table 6. The RDMs used in all the experiments correspond to different values of $R$. This further shows that having RDMs at different information rate $R$ is key to improving performance.

In Table 5, we study the affect of number of RDMs on distilling CLIP like models. To reduce compute, we use a smaller teacher of size ViT-B-16 trained by [58]. We see that increasing number of RDMs leads to improved performance. We also provide results from [20] for comaprison.

**Analyzing CIFD gains with large teacher student capacity gap:** Here,

Table 6: Ablation study of number of RDMs and IBM. IBM is crucial when number of RDMs increases.

| Dataset | # RDMs | w/ IBM | w/o IBM |
|---|---|---|---|
| CIFAR-100 | No RDM | **45.68** | 44.76 |
| | 1 RDM | **46.17** | 45.34 |
| | 3 RDM | **46.68** | 46.04 |
| | 5 RDM | **47.25** | 45.14 |
| Imagenet (RN34 to RN18) | 1 RDM | **72.05** | 71.83 |
| | 3 RDM | **72.32** | 72.22 |

we study the performance of CIFD when the teacher student gap is increased. In Table 7 we see that as the size of the teacher (and correspondingly its performance) increases, the student performance also increases. In fact, the increase is montonic w.r.t. the teacher size. We also compare with the performance of DistKD [5] which also studied a similar premise. We find that not only does our method outperform DistKD, that unlike DistKD the proposed CIFD does not show drop in performance when the teacher size is increased. This indicates the robustness of our proposed method. The parameter ratio of teacher to student ranges from 1.86 to 5.12. We also study CLIP like models where the maximum parameter ratio is a larger 6.9 in Appendix B.4. We see that for 3 out of the 5 zero-shot classification datasets CIFD yields larger improvements over the next best competitor of ClipKD [20] when the capacity gap is larger. In zero-shot image-text retrieval, CIFD almost always yields larger improvements over the ClipKD when the capacity gap is larger. This indicates that CIFD excels when the teacher-student capacity gap is large.

**Training cost analysis:** Although, prior works on Teacher Assistants like [2, 3] showed promising results, training Teacher Assistants led to prohibitive increase in complexity as seen in Figure 1(b) where TA based methods are $3.5\times$ and $4.5\times$ more expensive than [1] when distilling from ResNet34 to ResNet18 on Imagenet. We computed these numbers based on

Table 7: Acc. (%) on ImageNet when distilling increasingly larger teachers into ResNet18.

| Student | Teacher | DistKD [5] | CIFD |
|---|---|---|---|
| | | **Top-1/Top-5** | **Top-1/Top-5** |
| ResNet18 | ResNet34 | 72.07/90.42 | **72.32/90.88** |
| ResNet18 | ResNet50 | 72.12/— | **72.40/91.01** |
| ResNet18 | ResNet101 | 72.08/— | **72.48/91.16** |
| ResNet18 | ResNet152 | 72.24/— | **72.60/91.17** |

the number of Multiply and Accumulations (MACs) for every forward pass, more details in Appendix D.2. On the otherhand, our method is only $1.08\times$ more expensive than [1, 4, 57, 6, 5], thus bringing training cost of TA based methods close to other current innovations. For the CIFAR-100 experiments in Table 1, TAKD [2] is $2\times$ more expensive and DGKD [3] is $5\times$ more expensive than proposed.

## 5   Conclusion

In this paper we present a novel knowledge distillation framework called the Controlled Information Flow (CIFD). CIFD consists of two main components. The first is a lightweight Rate Distortion Module (RDM) that replaces the expensive Teacher Assistants by using the teacher model's embedding and a noisy communication channel. Importantly, since the RDM uses the teacher model's input embeddings, it does not have to relearn low level feature extractors, thus making the model significantly smaller than a TA. Second, we propose an Information Bottleneck Module to prevent the student model from overfitting in the presence of the teacher and multiple RDMs. The resulting framework shows impressive performance on Imagenet and significantly outperforms CLIP specific distillation methods on CLIP models. Finally, we corrobrated that an increased number of RDMs with diverse Rs is a key factor for better distillation, entailing only a small increase in computation.

**Limitations and impact:** An interesting direction is to study alternative algorithmic formulations for using the information from RDMs, i.e., sequentially enabling and disabling which RDM give feedback at what points of training. Regarding societal impact, the student model depends on the teacher model to transfer concepts. Unfortunately, this also means biases present in the teacher due to its training data are also transferred to the student.

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

# A Implementation Details

## A.1 CIFAR-100

### A.1.1 Simple CNNs (studied in Table 1)

Table 8: CIFAR-100 Model architecture for simple CNNs. 'C' stands for convolution layer with kernel size 3, stride 1, and padding 1. The number following it indicates the number of filters. 'FC' stands for fully connected layer with the number following it indicating the number of hidden neurons. 'MP' stands for MaxPool. After every convolutional or fully connected layer (except the output layer), we have a ReLU activation.

| Model | Architecture |
|---|---|
| 10 layer CNN | C32, C32, MP, C64, C64, MP, C128, C128, MP, C256, C256, C256, C256, MP, FC512, FC100 |
| 8 layer CNN | C32, C32, MP, C64, C64, MP, C128, C128, MP, C256, C256,MP, FC64, FC100 |
| 6 layer CNN | C32, C32, MP, C64, C64, MP,C128, C128 , FC256, FC100 |
| 4 layer CNN | C32, C32, MP, C64, C64, MP, FC256, FC100 |
| 2 layer CNN | C32, MP, C32, MP, FC256, FC100 |

Table 8 shows the CNN architectures used for the experiments. The output of FC512 from the 10-layer CNN is taken as the teacher's image embedding and passed to the RDM module. The RDM module consists of three hidden layers FC512, FC306, FC100, followed by an output layer of FC100. A bottleneck based on the design of [24] is placed after FC306. The bottleneck adds noise and computes the probability $\log q(\hat{Y})$ as shown in (4). During inference, i.e., when the RDM is not being trained, for stability, we quantize the representation to integers instead of adding uniform noise. Note that since the uniform noise during training is sampled from $[-0.5, 0.5]$, statistically, the two are equivalent.

The IBM module is only used in the two layer CNN, the student model. Unlike the RDM, in the IBM we do not design a dedicated encoder or decoder, we allow the preceding and succeeding layers of the model itself to act like the encoder and decoder respectively. The IBM just consists of a layer that adds uniform random noise in the range $[-0.5, 0.5]$. During inference the noise addition is disabled.

The student model is trained using the loss function (9). We used the Optuna algorithm [59] along with the Asynchronous Hyperband Scheduler [60] in the RayTune package [61] package for hyperparameter optimization. Using this package we optimized, distillation temperature ($\tau$), learning rate of the optimizer (SGD), learning rate decay, momentum of the optimizer, dropout (this is the KD dropout proposed in [46]), and all the $\lambda$s ($\lambda_{KL}, \lambda_1, \ldots, \lambda_5, \lambda_{IBM}$) involved in (9). The weight decay was fixed to $10^{-4}$. We did the same tuning for both DGKD [3] and TAKD [2]. The RDM model is trained using the loss function (5), without the reconstruction losses. We did similar hyperparamter tuning as above for TAKD [2]and DGKD [3].

### A.1.2 Competitive CNNs (studied in Table 2)

In this case, the architectures are standard and we follow it from [4]. The RDM module is a fully connected network with three hidden layers with the bottleneck after the second hidden layer. We train three RDMs with $R = 1.0, 0.8, 0.6$ for 30 epochs.

For training student models, we train with SGD for 240 epochs. Starting learning rate is 0.05 and decayed by a factor of 0.1 at 150, 160, and 180 epochs. We set $\lambda_{CE} = \lambda_{KL} = \lambda_1 = \lambda_2 = \lambda_3 = 1$, where $n \in 1, ..., 3$ indexes the three RDMs and KD dropout was set to 0.25. We set $\tau = 2.0$. The weight of the embedding loss (set to 100.0). We explored $\lambda_I \in \{0.001, 0.005, 0.01\}$ and set it to 0.005.

Unlike the RDM, in the IBM we do not design a dedicated encoder or decoder, we allow the preceding and succeeding layers of the model itself to act like the encoder and decoder respectively. The IBM just consists of a layer that adds uniform random noise in the range $[-0.5, 0.5]$. During inference the noise addition is disabled.

## A.2 ImageNet

For the experiments related to ImageNet we used the standard model architectures. Similar to CIFAR-100, the RDM module is a fully connected network with three hidden layers with a bottleneck after the second hidden layer. The teacher embedding is accessed after the avgpool. For RDMs for the ResNet34 model, there are three hidden layers of size 512, with activation after the first and third layers only. The embedding is linearly transformed to a logits of a 1000 class classifier to compute the other components in (9). For the ResNet50 RDM, instead of hidden neuron size of 512, we had 1024. The bottleneck follows the mechanism [24]. Instead of MSE loss, the bottleneck is trained with the smooth L1 loss in (5). We split the training data in imagenet into a training and validation set with ratio $0.95 : 0.05$. We train the RDMs for 30 epochs and monitor the classification accuracy on this validation set. The values of $R$ are $10^4, 5 \times 10^3, 10^3$.

For training the student model, in (9), we set, $\lambda_{CE} = \lambda_{KL} = \lambda_1 = \lambda_2 = \lambda_3 = 1$, where $n \in 1, ..., 3$ indexes the three RDMs and KD dropout was set to $0.25$. We set $\tau = 2.0$. Instead of MSE loss for embedding reconstruction, we use the smooth $L_1$ loss. We search for only two hyperparameters of $\lambda_I$ (set to $0.001$) and the weight of the embedding loss (set to $100.0$). We set the learning rate to $2.0$ for the SGD optimizer, batch size of 1024 per GPU (4 GPUs), and weight decay to $5 \times 10^{-5}$. We train the system for 88 epochs. For all other hyperparameters, we used the defaults from [62]. We split the training data with a ratio of $0.95 : 0.05$ and used the smaller part as validation for hyperparameter tuning.

Unlike the RDM, in the IBM we do not design a dedicated encoder or decoder, we allow the preceding and succeeding layers of the model itself to act like the encoder and decoder respectively. The IBM just consists of a layer that adds uniform random noise in the range $[-0.5, 0.5]$. During inference the noise addition is disabled.

## A.3 CLIP

Table 9: CLIP model architectures. The architecture corresponds to the implementation in [58].

| Model Tag | Embedding Dimension | Vision Encoder | | | | | Text Encoder | | | | | | Total Params |
|---|---|---|---|---|---|---|---|---|---|---|---|---|---|
| | | Model arch | Image Size | Layers | Width | Patch Size | Model arch | Context length | Vocabulary size | Width | Heads | Layers | |
| RN50 | 1024 | ResNet50 | 224 | [3, 4, 6, 3] | 64 | N/A | Transformer | 77 | 49408 | 512 | 8 | 12 | 106M |
| ViT-S-16 | 384 | ViT | 224 | 12 | 384 | 16 | Transformer | 77 | 49408 | 384 | 6 | 12 | 62M |
| ViT-B-16 | 512 | ViT | 224 | 12 | 768 | 16 | Transformer | 77 | 49408 | 512 | 8 | 12 | 154M |
| ViT-L-14 | 768 | ViT | 224 | 24 | 1024 | 14 | Transformer | 77 | 49408 | 768 | 12 | 12 | 428M |

We use the package [58] for implementation. We implemented and ran the experiments for the works of both TinyClip [19] and CLIPKD [20]. For TinyClip we only used the Knowledge Distillation part of the proposed idea and not the pruning methodology as pruning is orthogonal to Knowledge Distillation and can be combined with any of the proposed models here. We used a batch size of 1024 (256 per GPU), and a learning rate of $10^{-3}$. For TinyClip we got better results from using a larger batch size of 6000. The models are trained for 32 epochs, with 3 million image-text pairs per epoch. Warmup is set to around 1/6th of the training. For hyperparameter tuning we use a subset of the CC12 dataset [47]. Hyper-parameters for CLIP involve only two $\lambda_I$ and $\lambda_{CL}$. We tried the $\lambda_{CL}$ suggested value from (Yang et al., 2023) of $0.0005$ and another $0.005$, we chose the latter. For $\lambda_I$, we tested four values and selected the best based on validation performance. All hyper-parameter tests were restricted to the RN-50 architecture. For all other architectures, the same hyper-parameter settings were used, unchanged.

The RDM training in CLIP models is done for seven epochs, each epoch consisting of 3 million image-text pairs. We use the same settings as training the student model.

# B More experimental results

## B.1 CIFAR-100

Table 10 compares the results on CIFAR-100 dataset with more baselines. As we can see, our proposed method achieves competitve performance on this dataset.

Table 10: Top-1 accuracies (%) on CIFAR-100.

| Teacher | WRN-40-2 | resnet32x4 |
|---|---|---|
| Student | WRN-40-1 | resnet8x4 |
| Teacher | 75.61 | 79.42 |
| Student | 71.98 | 72.50 |
| AT [54] | 72.8 | 73.4 |
| SP [63] | 72.4 | 72.9 |
| VID [8] | 73.3 | 73.1 |
| RKD [64] | 72.2 | 71.9 |
| PKT [65] | 73.5 | 73.6 |
| AB [66] | 72.4 | 73.2 |
| FT [14] | 71.6 | 72.9 |
| NST [67] | 72.2 | 73.3 |
| KD [1] | 73.5 | 73.3 |
| CRD [55] | 74.1 | 75.5 |
| WSLD* [56] | 73.7 | 74.8 |
| IPWD [4] | **74.6** | **76.0** |
| **CIFD** | **74.6** | **76.0** |

## B.2 ImageNet

Table 11 shows more baselines for experiments on ImageNet. As we can see our proposed method outperforms all the compared methods.

## B.3 Zero-shot results for CLIP distilled models

Table 14 shows the zero-shot image text classification results using CLIP like models over five datasets. Table 15 shows the zero-shot image text retrieval results over COCO and FlickR30k test sets. Across all classification and retrieval, our model **consistently outperforms** CLIP spcific distillation methods. This showcases the generality of the idea proposed. Further, we see the largest gains due to the proposed method when the gap between the teacher and student size is the largest.

## B.4 Studying the effect of student-teacher capacity gap in CLIP distilled models

We study the effect of large teacher student capacity gaps in CLIP like models. In Table 12 and Table 13, we compare the difference when a ViT-L-14 teacher was used to train ViT-B-16 and ViT-S-16 students. The parameter ratios are 2.8 and 6.9, respectively. CIFD shows greater improvement over baseline when the teacher student ratio is large for 3 out of the 5 zero-shot classification datasets and almost always for the two zero-shot retrieval datasets.

## C Connections between IBM and Masked Image Modeling

Masked Image Modeling (MIM), as shown in Figure 6a, has had major success in pretraining large image models like BEiT[16]. Here, $T$ is used to represent the input, $T_M$ is used to represent the masked input, $U$ is the tokens of the image corresponding to the masked out parts, and $\hat{U}$ is their predicted value as predicted the MIM model. In the BEiT variant of MIM, an image is first converted into tokens using a pretrained tokenizer (like a discrete VAE). Next, a masked version of the image ($T_M$) is fed into the MIM encoder which attempts to predict the tokens of the masked parts. We can write the MIM training objective as

$$\min -\mathbb{E}_{U,\hat{U}} \left[ \log p(U|\hat{U}) \right] \qquad (14)$$

Masked Generative Distillation (MGD), as shown in Figure 6b, is an extension of using MIM type of training for distillation. Here, $T$ is used to represent the input, $U$ is the teacher model embedding corresponding to the full image, and $\hat{U}$ is their predicted value as predicted the MIM model. In MIM,

Table 11: Acc. (%) on ImageNet.

| | Same arch. style | | Diff. arch. style | |
| Teacher
Student | ResNet-34
ResNet-18 | | ResNet-50
MobileNet-v1 | |
| | **Top-1** | **Top-5** | **Top-1** | **Top-5** |
|---|---|---|---|---|
| Teacher | 73.31 | 91.42 | 76.16 | 92.87 |
| Student | 69.75 | 89.07 | 68.87 | 88.76 |
| AT [54] | 71.03 | 90.04 | 70.18 | 89.68 |
| NST [67] | 70.29 | 89.53 | — | — |
| RKD [64] | 70.40 | 89.78 | 68.50 | 88.32 |
| Online KD [68] | 70.55 | 89.59 | — | — |
| FSP [69] | 70.58 | 89.61 | — | — |
| SP [63] | 70.62 | 89.80 | — | — |
| AT [54] | 71.03 | 90.04 | 70.18 | 89.68 |
| Overhaul [7] | 71.03 | 90.15 | 71.33 | 90.33 |
| CRD [55] | 71.17 | 90.13 | 69.07 | 88.94 |
| KD [1] | 70.67 | 90.04 | 70.49 | 89.92 |
| FT [14] | 71.43 | 90.29 | — | — |
| SSKD [70] | 71.62 | 90.67 | — | — |
| DKD [71] | 71.70 | 90.41 | 72.05 | 91.05 |
| Residual KD [13] | 71.79 | 90.25 | — | — |
| TAKD [2] | 71.37 | 90.27 | — | — |
| DGKD [3] | 71.73 | 90.82 | — | — |
| IPWD [4] | 71.88 | 90.50 | 72.65 | 91.08 |
| CD [72] | 71.90 | 90.70 | — | — |
| IFD [6] | 71.94 | 90.68 | 73.16 | 91.24 |
| NormKD [57] | 72.03 | 90.64 | 72.79 | 91.08 |
| WSLD [56] | 72.04 | 90.70 | 71.52 | 90.34 |
| DistKD [5] | 72.07 | 90.42 | — | — |
| MGD [15] | 71.80 | 90.40 | 72.59 | 90.94 |
| MLKD [73] | 71.90 | 90.55 | 73.01 | 91.42 |
| CTKD [74] | 71.51 | 90.47 | — | — |
| LSKD [75] | 72.08 | 90.74 | 73.22 | 91.59 |
| DiffKD [11] | 72.22 | 90.64 | **73.62** | 91.34 |
| CIFD | **72.32** | **90.88** | 73.51 | **91.74** |

the target for prediction are the tokens corresponding to the masked parts of the input. In the case of MGD, the student model is tasked to predict the teacher model's embedding of the original image. MGD has shown promising results in distillation. The loss corresponding to the Masked Generative Distillation is written as

$$\min - \mathbb{E}_{U,\hat{U}} \left[ \log p(U|\hat{U}) \right].$$ (15)

The loss functions rightfully appear the same. However, the process differs in two key places. First is where the masking is applied. In MIM, the masking is done on the input. In MGD, the masking is done after student backbone preprocessing. Secondly, the likelihood maximization formulation is different, i.e., in MIM, the tokens are discrete classes so a cross-entropy loss is used, whereas in MGD, the teacher embedding is continuous so MSE is used. Thus the likelihood distribution in MIM is multinomial whereas in MGD it is gaussian. Note, for MGD we are only studying the masked distillation loss proposed. The other loss function is not relevant here.

Before comparing with IBP we present a small lemma that minimizing the negative log-likelihood is the same as minimizing the negative of the Mutual Information between two variables.

**Lemma 1.** *Minimizing the negative log-likelihood of predicting a non-learnable $U$ from some learnable $\hat{U}$*

$$\min - \mathbb{E}_{U,\hat{U}} \left[ \log p(U|\hat{U}) \right].$$ (16)

*is equivalent to*

$$\min - I(U; \hat{U}).$$ (17)

Table 12: Zero-shot image classification performance with same teacher different students. Larger the parameter ratio between teacher to student, CIFD shows larger benefit over CLIPKD for 3 out 5 datasets.

| Method | Model | Param ratio | IN(%) | | IN-V2(%) | | IN-R(%) | | IN-A(%) | | ObjNet(%) | |
|---|---|---|---|---|---|---|---|---|---|---|---|---|
| | | | Top-1 | Top-5 | Top-1 | Top-5 | Top-1 | Top-5 | Top-1 | Top-5 | Top-1 | Top-5 |
| Teacher | ViT-L-14 | | 79.22 | 95.52 | 72.05 | 92.22 | 90.85 | 97.80 | 69.07 | 89.40 | 72.80 | 90.01 |
| CLIPKD [20] | ViT-S-16 | 6.9 | 39.42 | 69.71 | 33.83 | 63.54 | 45.43 | 70.50 | 9.21 | 29.17 | 22.11 | 45.45 |
| CIFD (1 RDM) | ViT-S-16 | 6.9 | 42.79 | 73.06 | 37.14 | 66.75 | 50.75 | 75.80 | 10.31 | 32.40 | 23.11 | 46.98 |
| Δ (CIFD-CLIPKD) | ViT-S-16 | 6.9 | **3.37** | **3.35** | **3.31** | **3.21** | **5.32** | **5.3** | 1.1 | 3.23 | 1.0 | 1.53 |
| CLIPKD [20] | ViT-B-16 | 2.8 | 51.25 | 79.91 | 44.81 | 73.40 | 61.92 | 82.19 | 15.56 | 41.27 | 29.63 | 53.44 |
| CIFD (1 RDM) | ViT-B-16 | 2.8 | 54.10 | 81.94 | 47.36 | 76.24 | 65.55 | 85.26 | 17.69 | 45.19 | 32.63 | 57.14 |
| Δ (CIFD-CLIPKD) | ViT-B-16 | 2.8 | 2.85 | 2.03 | 2.55 | 2.84 | 3.63 | 3.07 | **2.13** | **3.92** | **3.0** | **3.7** |

Table 13: Zero-shot image-text retrieval performance on COCO [52] and FlickR30k [53] test sets. Larger the parameter ratio between teacher to student, CIFD shows larger benefit over CLIPKD almost always.

| Method | Model | Param ratio | COCO | | | | | | Flickr30k | | | | | |
|---|---|---|---|---|---|---|---|---|---|---|---|---|---|---|
| | | | Image → Text | | | Text → Image | | | Image → Text | | | Text → Image | | |
| | | | R@1 | R@5 | R@10 | R@1 | R@5 | R@10 | R@1 | R@5 | R@10 | R@1 | R@5 | R@10 |
| Teacher | ViT-L-14 | | 63.60 | 84.90 | 90.84 | 44.91 | 70.20 | 79.00 | 88.30 | 98.70 | 99.80 | 7386 | 91.92 | 95.44 |
| CLIPKD [20] | ViT-S-16 | 6.9 | 31.04 | 56.76 | 69.10 | 23.10 | 46.89 | 59.18 | 54.70 | 81.20 | 88.30 | 46.26 | 73.96 | 82.68 |
| CIFD (1 RDM) | ViT-S-16 | 6.9 | 37.70 | 64.84 | 76.12 | 25.29 | 49.73 | 61.17 | 62.00 | 85.50 | 91.50 | 50.36 | 76.70 | 84.58 |
| Δ (CIFD-CLIPKD) | ViT-S-16 | 6.9 | **6.66** | **8.08** | **7.02** | 2.19 | **2.84** | **1.99** | **7.3** | **4.3** | **3.2** | 4.1 | **2.74** | **1.9** |
| CLIPKD [20] | ViT-B-16 | 2.8 | 39.44 | 66.64 | 77.62 | 28.78 | 54.53 | 66.42 | 69.50 | 89.50 | 93.90 | 55.94 | 82.42 | 88.56 |
| CIFD (1 RDM) | ViT-B-16 | 2.8 | 44.90 | 71.72 | 80.66 | 31.64 | 56.73 | 67.49 | 73.60 | 91.80 | 96.00 | 60.10 | 84.36 | 90.00 |
| Δ (CIFD-CLIPKD) | ViT-B-16 | 2.8 | 5.46 | 5.08 | 3.04 | **2.86** | 2.2 | 1.07 | 4.1 | 2.3 | 2.1 | **4.16** | 1.94 | 1.44 |

Table 14: Zero-shot image classification performance. For ease we identify models based on their image encoder configuration. The full architecture details are given in Table 9. All methods use the first Teacher, MobileCLIP additionally uses Teacher-2[†].

| Method | Model | IN(%) | | IN-V2(%) | | IN-R(%) | | IN-A(%) | | ObjNet(%) | |
|---|---|---|---|---|---|---|---|---|---|---|---|
| | | Top-1 | Top-5 | Top-1 | Top-5 | Top-1 | Top-5 | Top-1 | Top-5 | Top-1 | Top-5 |
| Teacher | ViT-L-14 | 79.22 | 95.52 | 72.05 | 92.22 | 90.85 | 97.80 | 69.07 | 89.40 | 72.80 | 90.01 |
| Teacher-2[†] | ViT-L-14 | 75.54 | 94.58 | 69.84 | 90.89 | 87.59 | 97.08 | 70.49 | 90.87 | 66.00 | 86.00 |
| Student | RN50 | 36.47 | 64.40 | 30.96 | 57.69 | 42.02 | 68.20 | 7.08 | 26.83 | 20.99 | 44.98 |
| OpenCLIP [76] | RN50 | 34.57 | 61.97 | 30.08 | 57.48 | 42.36 | 67.97 | 7.25 | 25.12 | 21.33 | 43.44 |
| MobileCLIP[†] [23] | RN50 | 31.09 | 57.96 | 26.89 | 52.76 | 37.24 | 63.15 | 7.12 | 24.45 | 19.00 | 41.00 |
| TinyCLIP [19] | RN50 | 36.61 | 64.33 | 32.05 | 59.15 | 45.53 | 71.00 | 8.16 | 27.24 | 21.86 | 44.02 |
| CLIPKD [20] | RN50 | 46.32 | 75.77 | 40.77 | 70.98 | 53.88 | **76.62** | 11.51 | 34.28 | 27.54 | 51.66 |
| CIFD (1 RDM) | RN50 | **47.36** | **76.92** | **41.67** | **72.15** | **53.89** | 76.19 | **12.35** | **36.96** | **28.85** | **53.46** |
| MobileCLIP[†] [23] | ViT-S-16 | 28.93 | 54.82 | 24.86 | 49.41 | 33.11 | 58.54 | 5.75 | 21.61 | 17.00 | 37.00 |
| TinyCLIP [19] | ViT-S-16 | 31.29 | 58.01 | 26.30 | 52.25 | 37.03 | 62.87 | 5.69 | 22.44 | 16.43 | 36.90 |
| CLIPKD [20] | ViT-S-16 | 39.42 | 69.71 | 33.83 | 63.54 | 45.43 | 70.50 | 9.21 | 29.17 | 22.11 | 45.45 |
| CIFD (1 RDM) | ViT-S-16 | **42.79** | **73.06** | **37.14** | **66.75** | **50.75** | **75.80** | **10.31** | **32.40** | **23.11** | **46.98** |
| MobileCLIP[†] [23] | ViT-B-16 | 31.38 | 58.32 | 26.48 | 52.41 | 37.72 | 63.52 | 6.51 | 22.79 | 18.00 | 39.00 |
| CLIPKD [20] | ViT-B-16 | 51.25 | 79.91 | 44.81 | 73.40 | 61.92 | 82.19 | 15.56 | 41.27 | 29.63 | 53.44 |
| CIFD (1 RDM) | ViT-B-16 | **54.10** | **81.94** | **47.36** | **76.24** | **65.55** | **85.26** | **17.69** | **45.19** | **32.63** | **57.14** |

*Proof.* $-I(U; \hat{U}) = -H(U) + H(U|\hat{U}) = -H(U) - \mathbb{E}_{U,\hat{U}}\left[\log p(U|\hat{U})\right]$ which follow from definitions of Mutual Information and conditional entropy [34]. Since $U$ is not optimizable, $H(U)$ is a constant w.r.t. optimization. Thus the proof follows. □

Table 15: Zero-shot image-text retrieval performance on COCO [52] and FlickR30k [53] test sets. All methods use the first Teacher, MobileCLIP additionally uses Teacher-2[†].

| Method | Model | COCO | | | | | | Flickr30k | | | | | |
| | | Image → Text | | | Text → Image | | | Image → Text | | | Text → Image | | |
| | | R@1 | R@5 | R@10 | R@1 | R@5 | R@10 | R@1 | R@5 | R@10 | R@1 | R@5 | R@10 |
|---|---|---|---|---|---|---|---|---|---|---|---|---|---|
| Teacher | ViT-L-14 | 63.60 | 84.90 | 90.84 | 44.91 | 7020 | 79.00 | 88.30 | 98.70 | 99.80 | 7386 | 91.92 | 95.44 |
| Teacher-2[†] | ViT-L-14 | 55.88 | 79.68 | 86.62 | 35.86 | 60.13 | 70.21 | 85.90 | 97.60 | 99.20 | 65.88 | 87.72 | 92.78 |
| Student | RN50 | 33.44 | 59.80 | 70.96 | 22.91 | 47.18 | 59.03 | 59.50 | 83.90 | 89.60 | 45.52 | 73.44 | 81.98 |
| OpenCLIP [58] | RN50 | 31.48 | 56.98 | 68.06 | 19.83 | 43.30 | 54.74 | 52.10 | 79.20 | 86.40 | 40.30 | 66.62 | 76.44 |
| MobileCLIP[†] [23] | RN50 | 28.82 | 55.26 | 66.16 | 18.34 | 40.17 | 52.36 | 48.20 | 74.40 | 83.70 | 36.74 | 63.58 | 73.84 |
| TinyCLIP [19] | RN50 | 35.38 | 60.66 | 72.02 | 21.97 | 45.78 | 57.42 | 59.50 | 83.20 | 88.60 | 43.08 | 70.58 | 80.26 |
| CLIPKD [20] | RN50 | 38.24 | 63.94 | 74.56 | 25.98 | 50.47 | 62.06 | 61.90 | 85.80 | 91.10 | 49.92 | 77.80 | 85.62 |
| CIFD (1 RDM) | RN50 | **40.70** | **66.78** | **76.64** | **26.02** | **50.90** | **62.50** | **65.40** | **87.60** | **93.10** | **50.46** | **78.34** | **85.64** |
| MobileCLIP[†] [23] | ViT-S-16 | 28.08 | 54.70 | 66.00 | 17.59 | 39.46 | 51.48 | 48.60 | 75.50 | 84.40 | 35.92 | 63.38 | 73.94 |
| TinyCLIP [19] | ViT-S-16 | 29.06 | 54.78 | 66.52 | 17.95 | 40.19 | 52.02 | 50.40 | 74.90 | 84.50 | 36.74 | 63.60 | 74.04 |
| CLIPKD [20] | ViT-S-16 | 31.04 | 56.76 | 69.10 | 23.10 | 46.89 | 59.18 | 54.70 | 81.20 | 88.30 | 46.26 | 73.96 | 82.68 |
| CIFD (1 RDM) | ViT-S-16 | **37.70** | **64.84** | **76.12** | **25.29** | **49.73** | **61.17** | **62.00** | **85.50** | **91.50** | **50.36** | **76.70** | **84.58** |
| MobileCLIP[†] [23] | ViT-B-16 | 30.80 | 55.80 | 67.50 | 19.38 | 41.71 | 54.01 | 53.20 | 79.80 | 88.30 | 39.72 | 67.24 | 77.54 |
| CLIPKD [20] | ViT-B-16 | 39.44 | 66.64 | 77.62 | 28.78 | 54.53 | 66.42 | 69.50 | 89.50 | 93.90 | 55.94 | 82.42 | 88.56 |
| CIFD (1 RDM) | ViT-B-16 | **44.90** | **71.72** | **80.66** | **31.64** | **56.73** | **67.49** | **73.60** | **91.80** | **96.00** | **60.10** | **84.36** | **90.00** |

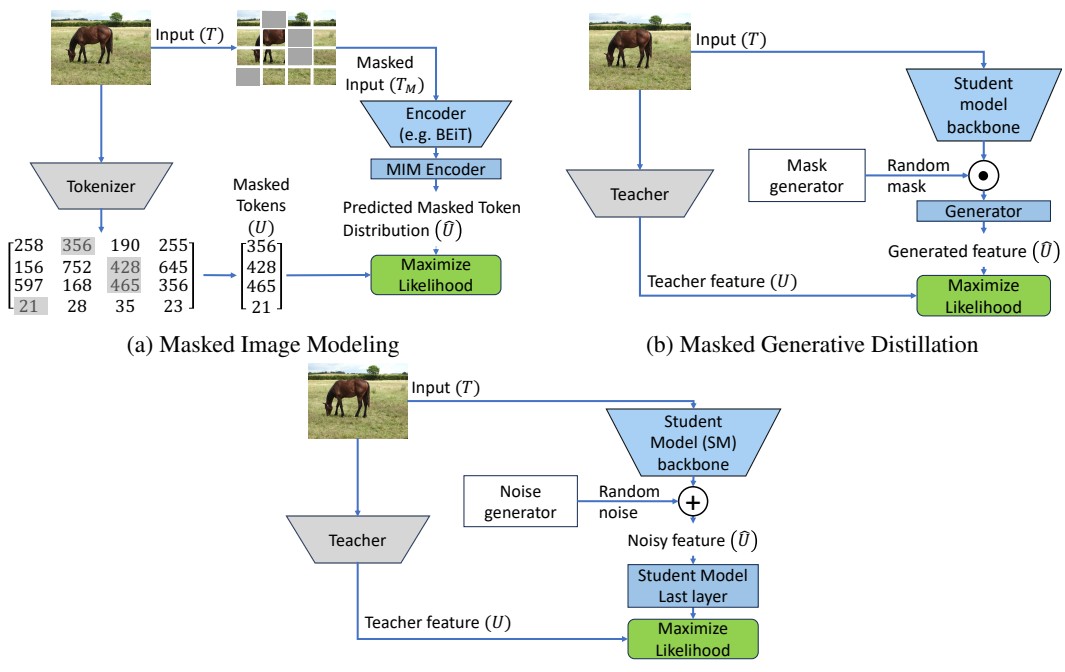

(a) Masked Image Modeling

(b) Masked Generative Distillation

(c) Information Bottleneck Module

Figure 6: Relation between Masked Image Modeling (MIM), Masked Generative Distillation (MGD), and Information Bottleneck Module (IBM) for Distillation.

The IBP objective function, applicable to all systems shown in Figure 6 is written as

$$\min -I(U; \hat{U}) + \lambda_I I(T; \hat{U}). \tag{18}$$

For simplicity of exposition we split Theorem 1 into the following lemmas one each for MIM and MGD respectively.

**Lemma 2.** *The objective function of MIM* (14) *is an upper bound on the objective function from the Information Bottleneck Principle* (18).

*Proof.* We start with the Information Bottleneck objective

$$-I(U; \hat{U}) + \lambda_I I(T; \hat{U}). \tag{19}$$

We note that $\hat{U} - T_M - T$ forms a Markov Chain. Thus leveraging the Data Processing Inequality [34], we note that $I(T; \hat{U}) \leq I(T; T_M)$. We note that $I(T; T_M)$ is constant w.r.t. the minimization, since it does not have any parameters that are optimized by gradient descent. Hence, we can drop that term. By leveraging Lemma 1, we can see that the MIM objective (14) is an upper-bound on the (18). $\qquad\square$

It is tempting to assume that MGD is also an upper-bound similar to MIM. Unfortunately, this is not the case in general. This is because the noise is added after backbone processing and this processing is optimizable. However, under certain conditions, we show that it is a valid upper-bound.

**Lemma 3.** *Let $\hat{U}_{-M}$ denote the random variable prior to the masking operation in MGD (Figure 6b). Then, the objective function of MGD* (15) *is an upper bound on the objective function from the Information Bottleneck Principle* (18) **if** *$\hat{U}_{-M}$ is a discrete random variable and the mapping from $T$ to $\hat{U}_{-M}$ is deterministic.*

*Proof.* We again start with the Information Bottleneck objective

$$-I(U; \hat{U}) + \lambda_I I(T; \hat{U}). \tag{20}$$

We note that $\hat{U} - \hat{U}_{-M} - T$ forms a Markov Chain. Thus leveraging the Data Processing Inequality [34], we note that $I(T; \hat{U}) \leq I(T; \hat{U}_{-M})$. Unlike the MIM case, $I(T; \hat{U}_{-M})$ is not a constant w.r.t. minimization. By definition of Mutual Information, we can write

$$I(T; \hat{U}_{-M}) = H(\hat{U}_{-M}) - H(\hat{U}_{-M}|T), \tag{21}$$

where by slight abuse of notation $H$ represents differential or discrete entropy based on if the random variable is continuous or discrete respectively. Since the mapping from $T$ to $\hat{U}_{-M}$ is deterministic, it follows that $H(\hat{U}_{-M}|T) = 0$. If we assume that $\hat{U}_{-M}$ is discrete, then $H(\hat{U}_{-M})$ is discrete entropy and we know that $H(\hat{U}_{-M}) \geq 0$ [34]. Thus, by dropping that term, we get an upper bound on the IBP objective. $\qquad\square$

**Remark 1.** *If $\hat{U}_{-M}$ is not discrete, then $H(\hat{U}_{-M}) < 0$ is possible, i.e., differential entropy can be negative. Thus, we cannot guarantee that the MGD objective is always an upper-bound of IBP in that case.*

**Remark 2.** *The MGD upper-bound employs two relaxations that make it an upper-bound on IBP. First is the Data Processing Inequality to show that $I(T; \hat{U}) \leq I(T; \hat{U}_{-M})$. The second (under the discrete assumption) that $H(\hat{U}_{-M}) \geq 0$.*

The loss function of our IBM is directly written as an approximation of the IBP objective as shown in Section 3.2. Unlike MIM or MGD, we do not drop the second term and instead use a non-parametric upper-bound to approximate it. This empirically should ensure a tighter approximation to the IBP objective than the other two. IBM works better because it forces the student to focus on those features necessary to predict the teacher embedding (first term) and remove information not useful in the prediction (second term), whereas in MIM and MGD the removal of information is not present. There have recently been works studying how IBP helps reduce generalization errors [17] which provides support to the proposed idea.

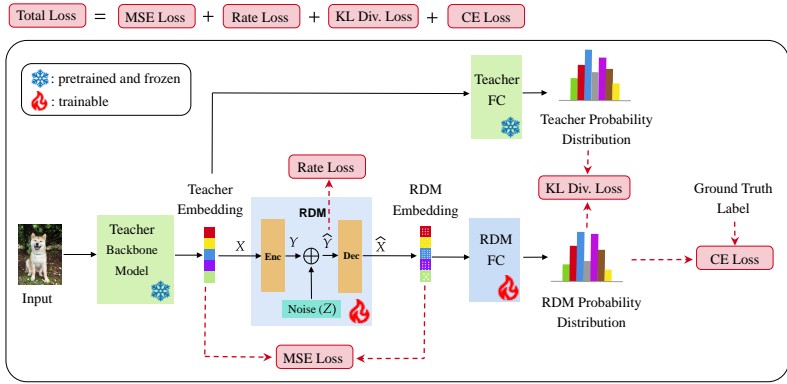

Figure 7: Training the RDM for classification

# D   More details

## D.1   Illustration of training RDMs and Student Models for classification tasks

In Figure 7 we show how the RDMs are trained for classification. This is the analogue of Figure 1(a) where there is no classification involved. In this setup, the input (image as an example here) is passed through the teacher backbone model. The resulting embedding is passed through the RDM encoder, subject to some noise ($Z$) and is reconstructed by the RDM decoder. The RDM in the case of classification is trained like a multi-task learning module, i.e., it is tasked with reconstructing both the input embedding and projecting the reconstructed embedding to perform classification. The RDM is trained in a similar fashion to a student model during knowledge distillation, i.e., there is feature distillation (reconstruct the teacher embedding), logit distillation to preserve the dark knowledge, and a supervised cross entropy loss component. The resulting loss function used to train the RDM can be written as (5) which we re-iterate here.

$$\mathcal{L}'_R = \mathbb{E}_{X,Z}\left[\lambda_{CE}\mathcal{L}_{CE}(V,\hat{V}) + \lambda_{KL}KL(\hat{V}_T||\hat{V}_{RDM}) + R\left\|X-\hat{X}\right\|_2^2 - \log\left(q(\hat{Y})\right)\right]. \quad (22)$$

In Figure 8 we show how the student models are trained for classification in the presence of an RDM and the IBM. For simplicity, we assume there is one RDM. The input image is passed through the teacher backbone model and the obtained embedding is passed through the RDM to get the reconstructed embedding from the RDM. The RDM then provides a predictive distribution, like a teaching assistant, and we can also get the teacher's predictive distribution. The input image is also passed through the student model and the IBM to obtain the student embedding. The student embedding is then subject to feature distillation, i.e., the loss between the student embedding and the teacher embedding, and the student embedding and the RDM embedding is computed. In the case where the student embedding's dimension does not match the teacher or the RDM embedding dimension, a small trainable projector network is used. Finally, the output predictive distribution of the student is subject to both the classification loss and the KL divergence losses w.r.t. the teacher's distribution and the RDM's distribution. So, this mechanism combines feature distillation and the output logit distillation along with the RDM. In the figure, we only show 1 RDM, however, it can easily be extend to multiple RDMs. The loss function used in this case is

$$\mathcal{L}'_{CIFD} = \mathbb{E}_{X,Z_1^N,Z_S}\left[\lambda_{CE}\mathcal{L}_{CE}(V,\hat{V}_S) + \lambda_{KL}KL(\hat{V}_T||\hat{V}_S) + \lambda_{KL}\sum_{n=1}^{n=N}\lambda_n KL(\hat{V}_n||\hat{V}_S)\right.$$
$$\left. + \|U_S-U_T\|_2^2 + \sum_{n=1}^{n=N}\lambda_n\|U_S-U_n\|_2^2 - \lambda_I\log r(\hat{W})\right]. \quad (23)$$

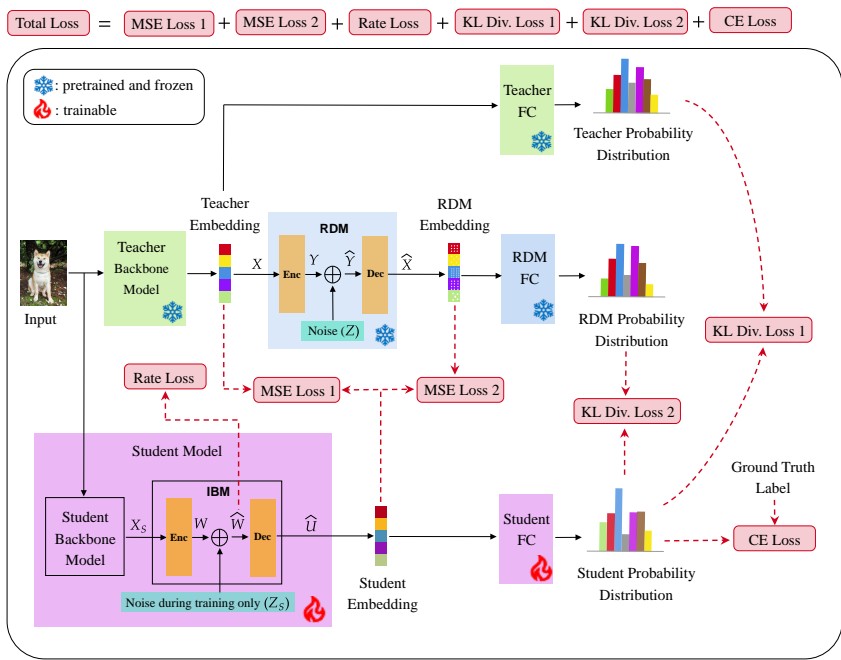

Figure 8: Training the Student Model for classification

## D.2 Computing training costs

Here, we explain how we compute the training cost for ResNet34 to ResNet18 distillation using CIFD. First, we compute the training costs of RDMs. The RDMs consist of a three layer fully Connected Network, with a computational complexity of $1.31$ MMACs (Mega or $10^6$ MACs) per image per forward pass. We train them for 30 epochs on $1.28 \times 10^6$ images for 30 epochs. Thus the cost of training three RDMs is $151$ TMACs (Terra or $10^{12}$ MACs). The significant chunk of computation in training RDMs is coming from running the teacher model in inference mode, which is $142$ PMACs (Peta or $10^{15}$ MACs). We can potentially extract the features from the teacher model once and reuse it for 30 epochs, in which case the training cost reduces to $4.7$ PMACs, but we do not consider that here.

The rest of the $619$ PMACs (out of the total $762$ PMACs), comes from the student model training. Note that the RDM forward passes are computationally insignificant at this stage, accounting for $0.38$ PMACs (out of the total $619$ PMACs). We do similar computations for [2, 3] while adding the computation for training one Teacher Assistant based on the training settings given in those papers.

Similarly we also compute the training cost for Knowledge Distillation [1]. The training costs of other high performing works [4, 6, 57, 5] are the same as Knowledge Distillation [1].

## D.3 Visualizing the Teacher and RDM embeddings

In Figure 9 we study the embeddings obtained from the teacher model and two of the RDM models trained on CIFAR-100 by plotting their tSNE [77]. The RDM in Figure 9b has a test accuracy of $52.32\%$ which is significantly closer to the teacher's accuracy than the other RDM in Figure 9c which has a test accuracy of $48.83\%$. First let us look at the orange class. The orange class which forms a concentrated cluster in the teacher and the more powerful RDM's embeddings, appears more scattered in the less accurate RDM's embeddings. The less concentrated cluster is easier for the student model to learn because coming up with a good representation that is sufficiently general to cover all examples in a class but at the same time not cover samples in another class is a hard problem. Thus, the RDM effectively bridges the gap by providing easier representation that the student can learn more easily before progressing to the harder representation. Further, we also notice that orange

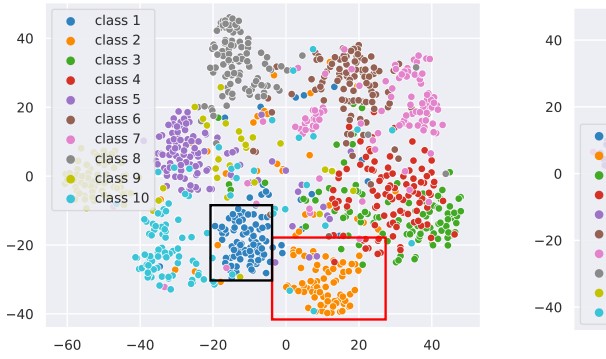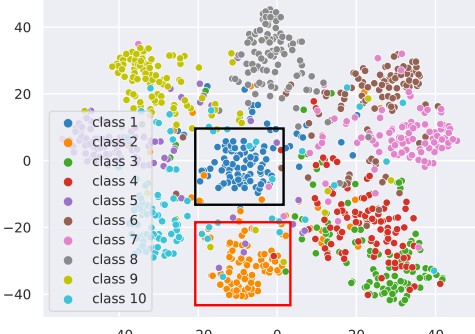

(a) tSNE of the Teacher model embeddings. Acc. 54.42 %.

(b) tSNE of the RDM model embeddings. Acc. 52.32 %.

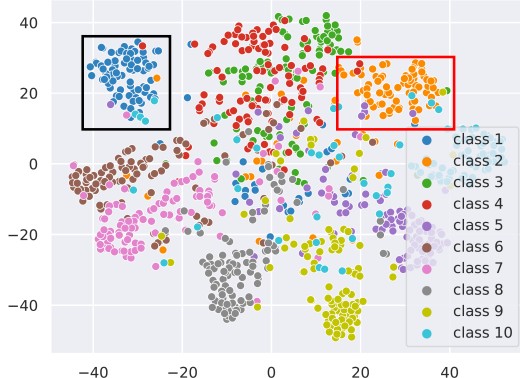

(c) tSNE of the RDM model embeddings. Acc. 48.83 %.

Figure 9: tSNE plots of embeddings for a subset of 10 classes of the CIFAR-100 dataset.

and blue clusters that are close to each other in both the teacher and the better performing RDM's embeddings are now farther apart in the poor performing RDM's embeddings.

### D.4 Details on CLIP baselines

While comparing with TinyCLIP [19], we disabled pruning, as pruning is orthogonal to our proposed idea. Instead we focused only on the Knowledge Distillation part from [19]. To do that in addition to the contrastive loss (11), we added the two terms as proposed in equations (1), Section 3.1 of [19]. These equations transfer the relative embedding distances from the teacher's embedding space to the student's embedding space.

