# OpenReview forum: "CIFD: Controlled Information Flow to Enhance Knowledge Distillation"
_NeurIPS.cc/2024/Conference — NeurIPS 2024 poster_

### Official Review · Reviewer_vQtp · 2024-07-09

**Soundness:** 2
**Presentation:** 3
**Contribution:** 3
**Rating:** 5
**Confidence:** 4

**Summary:**

Some existing methods alleviate the capacity gap between the teacher and student by setting up Teacher Assistants (TAs), introducing a large number of additional parameters and computational costs. Based on this, this paper proposes to train multiple RDM modules and connect multiple independent classification heads to generate branches with different performance to simulate the TA model. The authors think that this hierarchical model of extracting teacher knowledge can help alleviate the capacity gap between the teacher and student.

**Strengths:**

* Assistants and teachers sharing shallow modules are more efficient in terms of parameter quantity compared to multiple independent Assistants models.
* Although a large number of fully connected layers have been introduced, the proposed method hardly introduces any additional training overhead.

**Weaknesses:**

* I noticed that there is a significant difference in baseline performance between Table 1 and the original text, and Table 3 only uses a single Assistant for TAKD and DGKD, while the author's proposed method uses three RDM headers, which is not a fair comparison.
* There are significant differences in the value of R across different datasets, R=1 for CIFAR but R=10^4 for ImageNet. This means that the selection of hyperparameters on unfamiliar datasets is challenging, and the parameter tuning process may introduce multiple computational costs, which limits the versatility of the method.
* I noticed that the method proposed by the author introduces and trains at least $3N$ additional layers of MLP, and the forward process and loss calculation cost in distillation is also increased several times. However, the training cost is even the same as the method NormKD based solely on Logits Distillation without additional modules (Figure 1 (b)). Can you present the results with specific numerical values? What is the key to introducing so many parameters without introducing additional computational overhead?

**Questions:**

* The experiment was conducted when there was not much difference between the teacher and student models (common settings in current KD tasks). It cannot prove that the proposed method can alleviate capacity differences, as many methods have proven that such settings (e.g., Res34 $\to$ Res18) do not require the additional assistant model.
* Why use the penultimate layer feature of the teacher? Is this from theoretical analysis or empirical summary?
* For this paper, I think it would be better to place Related Works at the front.

**Limitations:**

The authors have discussed a limitation in Conclusion.

---

> ### Author Rebuttal · Authors · 2024-08-07
>
> We thank the reviewer for their time and effort. Detailed comments follow the summary.
>
> **Summary**
>
> - We showed that our proposed method outperforms TAKD and DGKD even when there is only 1 RDM. This ensures fairness. However, in terms of training cost, our method with 3 RDMs which is far superior is also far cheaper (as seen Fig. 1).
> - We clarified that the low cost of our method is due to the relatively cheap computation cost of RDMs compared to the student/teacher model, ability to train RDMs in parallel, and train them for lesser epochs. We also extended our numerical training cost analysis in appendix C.3 and provided more information.
> - We directly addressed the concern on the efficacy of CIFD over large-student teacher gap. We distilled ResNet18 using ResNet152, ResNet101, ResNet50 (Table 13) as teachers and showed that proposed CIFD shows increasing student performance with teacher size. We also analyzed CLIP models (Table 16, 17) and found that it provides more improvement over baseline when the teacher student capacity gap is large.
>
> **Detailed response**
>
> > Difference between TAKD, DGKD and reported numbers
>
> We wanted to clarify if the reviewer is referring to the difference in performance of TAKD and DGKD in Table 1 and the original numbers in their papers? If so, the problem is both those papers used the test set for parameter fine-tuning [A] and [B]. After rectifying these errors, we got the above numbers.
>
> [A] TAKD first author's comments on GitHub: \url{https://github.com/imirzadeh/Teacher-Assistant-Knowledge-Distillation/issues/19#issuecomment-732454350}
>
> [B] DGKD implementation by the first author. Validation function (defined in line 133-137) called in line 143, uses the test set to select the best accuracy across epochs in \url{https://github.com/wonchulSon/DGKD/blob/main/train.py}
>
> > Fairness between TAKD, DGKD, and CIFD
>
> Coming to the fairness between comparing TAKD and DGKD in Table 3, one fairness is along the axis of training cost. In which case, as seen in Fig. 1, we incur lower training cost despite using three RDMs and significantly outperform them. Further as seen below, even when comparing our 1 RDM results with theirs, our proposed method is better.
>
> ```
> | RN34 to RN18 on IN1k | Top-1 | Top-5 |
> +----------------------+-------+-------+
> |        TAKD          | 71.37 | 90.27 |
> |        DGKD          | 71.73 | 90.82 |
> |   Proposed (1 RDM)   | 72.05 | 90.70 |
> |   Proposed (3 RDMs)  | 72.32 | 90.88 |
> ```
>
>
> > Computation costs
>
> We would like to clarify that NormKD, DistKD, IPWD, IFD, and KD are indeed slightly lower cost than our method. However, our algorithm is on the pareto front of the training cost v/s performance curve.
>
> We use the number of MACs in the forward pass as a measure of computation. There are three reasons. Firstly, the number of MACs consumed by RDMs is insignificant compared to a forward pass of the teacher model during RDM training. For e.g., one forward pass of the RDM per image is $1.31 \times 10^6$ MACs (RDM for RN34), whereas RN34 forward pass is $3.7 \times 10^9$ MACs. Thus, during RDM training the teacher model's forward pass is the biggest cost. Secondly, the RDM is trained only for 30 epochs. Finally, the RDMs are trained in parallel amortizing the cost of the forward pass of the teacher model during training. These in total reduce the cost of the training. Below we give the cost computations (also presented in Appendix C.3).
>
> Our method in total costs 762 PMACs (Peta or $10^{15}$ MACs) for distilling RN18 from RN34. First, we compute the RDM training cost. RDMs consist of a three layer fully Connected Network, with a computational complexity of $1.31$ MMACs ($10^{6}$ MACs) per image per forward pass. A 30 epoch training results in total cost of $151$ TMACs ($10^{12}$ MACs) for three RDMs, the significant chunk of computation in training RDMs is coming from running the teacher model in inference mode ($142$ PMACs). The rest of the compute comes from the student model training when the three RDMs, student, and teacher are being used. Note that the RDM forward passes are computationally insignificant at this stage, only $0.38$ PMACs. Finally, we compute the training cost for existing methods like NormKD, DistKD, IPWD, IFD, and KD as $704$ PMACs, using the cost of the student and teacher models.
>
> > Larger teacher experiments
>
> We trained distilled RN18 from RN152, RN101, RN50 teachers. DistKD [5] showed that ResNet152, ResNet101 to ResNet18 faces the issue of large capacity gap (Table 3 of DistKD). The parameter ratio of from largest teacher to student is 5.12. Experimental results are in Table 13. First we observed that student performance increased with teacher size. Secondly, our results outperformed that of DistKD. This directly addresses the reviewer's concern about large teacher student capacity gap.
>
> Additionally, we also compare CLIP models. Due to resource constraints, we compared the improvement provided by proposed CIFD over the nearest baseline CLIPKD [31]. Specifically, we compared the difference when a ViT-L-14 teacher was used to train ViT-B-16 and ViT-S-16 students. The parameter ratios are 2.8 and 6.9, respectively. Results are in Table 16 and 17. CIFD shows greater improvement over baseline when the teacher student ratio is large for 3 out of the 5 zero-shot classification datasets and almost always for the two zero-shot retrieval datasets. The capacity gap in CLIP-like models is much larger than the capacity gap in traditional KD settings such as RN34-18 and RN50-MobileNet V1.
>
> > Layer for teacher embedding
>
> This is based on the idea that the penultimate layer output is usually considered the embedding of the network. Additionally, the embeddings holds all the important information for classification and we can easily remove that information using our RDM.
>
> > Related work position
>
> Thank you for the feedback, absolutely we will move it

---

> ### Comment · Reviewer_vQtp · 2024-08-13
>
> Thank you for providing a detailed response that resolved some of my doubts. The range of uncertainty in hyperparameter R is still one of my concerns, but experiments based on CLIP are indeed very distinctive. Based on the opinions of other reviewers and my understanding, I will increase the score to 5.

---

> ### Author Response · Authors · 2024-08-13
> **Thank you!**
>
> Thank you for your time and effort, we greatly appreciate your response. We also wanted to acknowledge your inputs in helping us improve the paper, thank you.
>
> We apologize for missing the point on the hyper-parameter R. From our analysis, we found that the values of R are inversely proportional to the capacity gap between the teacher and the student. Since $R^{-1}$ controls the weight of the bottleneck rate, too small an $R$ and the RDM will focus more on compression than retaining features crucial for accuracy, resulting in a poor RDM. Since the RN34 to RN18 gap is small (both in size and performance), smaller $R$ values were more appropriate and hence the selection. We plan to add this insight to our paper in the final version.
>
> If you have any further questions or concerns to improve our paper, please let us know and we are happy to discuss. Thank you again.

---

### Official Review · Reviewer_iLhT · 2024-07-14

**Soundness:** 3
**Presentation:** 3
**Contribution:** 3
**Rating:** 5
**Confidence:** 4

**Summary:**

Inspired by Shannon’s rate-distortion theory, this paper proposes two modules, namely the Rate-Distortion Module and the Information Bottleneck Module, to construct intermediate representations for knowledge distillation. Extensive experiments on various datasets demonstrate the effectiveness of this method.

**Strengths:**

1. This paper is well-presented and easy to understand.
2. This method not only works for traditional CNN networks but also performs well on modern CLIP models.
3. Extensive experiments demonstrate the effectiveness of this method.

**Weaknesses:**

1. The author's motivation and explanation for TA distillation are not very convincing. In my view, the RDM and IBM proposed in this work can be interpreted as two adapters connected to the teacher and the student respectively for distillation, and the principle is similar to the FT method.
2. In Table 2, 9 and 10, most of the compared methods were published in 2022 or before. The authors are encouraged to compare your method with recent state-of-the-art methods such as MLKD[1], CTKD[2], and LSKD[3].
3. In Eq. 5, I am a little confused about the author's formula representation. Generally speaking, the left side of the comma is the network to be trained, and the right side is the learning target. But the author seems to have it reversed here.

References:
[1]. Multi-Level Logit Distillation. CVPR 23.
[2]. Curriculum Temperature for Knowledge Distillation. AAAI 23.
[3]. Logit Standardization in Knowledge Distillation. CVPR 24.

**Questions:**

1. $q(\hat{Y})$ is not clearly marked in Fig. 2. Which part of the network produces it?
2. It would be more beneficial if the author could add a figure on how to perform CIFD distillation on the CLIP model.

**Limitations:**

Please refer to weaknesses.

---

> ### Author Rebuttal · Authors · 2024-08-07
>
> We thank the reviewer for their time and effort. Detailed comments follow the summary.
>
> **Summary**
>
> - By using one RDM without IBM, we showed that the key to our superior performance compared to Factor Transfer (FT) [28] is the principled loss function used to train the RDM. This is in addition to the multiple RDMs and IBM which have no equivalents in FT.
>
> - We showed superior performance against newer works like MLKD, CTKD, and LSKD.
>
> **Detailed response**
>
> > Motivation and comparison with FT [28]
>
> Before we respond let us quickly summarize the work of FT [28]. Instead of using teacher's logits for KD, FT proposed using teacher's embeddings. They transfer the knowledge using a paraphraser network with a convolutional autoencoder architecture. The paraphraser encoder outputs a vector with dimension that is $k$ times the dimension of its input and the decoder used to reconstruct the encoder's input, is discarded after training the encoder. Kim et. al. explored both dimensionality reduction ($k=0.5$) and expansion ($k=4$), and settled on $k=0.5$ as the best performing. During the training of the student model, they use a translater network, similar to the encoder of the paraphraser network and task it to map the student embeddings to the output domain of the paraphraser encoder. Below, we will highlight the main difference between FT and our method, and explain that this difference is crucial in obtaining better performance.
>
> While both paraphraser and our RDM aim to do compression, the mechanism of training the compression module is different. Kim et. al. train the paraphraser network like an autoencoder. The paraphraser training limits information by dimensionality reduction (when $k<1$). We train our network using the Rate-Distortion criterion, the mathematically principled way of performing lossy compression. The crucial difference is that the RDM limits the rate of information flow by minimizing the mutual information between the latent representation ($\hat{Y}$ in Fig. 2(a)) and the input ($X$) in addition to the reconstruction error (see the difference between the loss function in our eqn. (7) and FT's eqn. (1)). This training scheme makes a significant difference in performance as shown in Table 15. In the table, we compare our method with only one RDM with the work of FT, and we see that our method is superior, despite removing all our other innovations, including multiple RDMs, and IBM. Thus, it shows that the principled training of the RDM plays a significant role in improving the performance of the student network. As a further comparison, we also look at IFD [20], which uses an ensemble of three paraphraser networks. However, our method with 3 RDMs and no IBM still outperforms it. This shows that just relying on diversity of initialization of paraphrasers does not yield benefits. However, training in RDMs in a principled manner to focus on different levels of information (fine to coarse) helps train the student better.
>
>
> To summarize our differences w.r.t. FT
>
> 1. We propose a principled way of compressing the features from the teacher model using insights from Rate-Distortion theory. This principled training loss leads to performance gains even in a standalone manner (71.83 > 71.43).
>
> 2. We propose multiple RDMs to help student network learn at different compression ratios. This has no equivalent in Kim et. al.
>
> 3. We propose the use of IBM in the student network to control the information flow in the student during training. This has no equivalent in Kim et. al.
>
> While at a certain abstraction we can regard RDMs as adapters, the key here is the loss function used to train the adapters. As the new results show, the principled loss function is critical for good performance.
>
> > Comparing with MLKD, CTKD, and LSKD
>
> Thank you for pointing them out. Our performance is superior to MLDK, CTKD, and LSKD works as seen in Table 14.
>
> > How is $q(\hat{Y})$ computed?
>
> $q(\hat{Y})$ is not produced by a network. Instead it is produced using a non-parametric distribution approximation process proposed in \cite{balle2017end}. The method also computes an upper-bound on the entropy as detailed in \cite{balle2017end}.
>
>
> > W3 and Q2
>
> Thank you for the inputs, we will rectify them.

---

> > ### Author Response · Authors · 2024-08-13
> > **Thank you and summary of changes**
> >
> > We wanted to thank the reviewer again for their insightful questions that have helped us improve the paper. We wanted to quickly summarize our previous response.
> >
> > - By using one RDM without IBM, we showed that the key to our superior performance compared to Factor Transfer (FT) [28] is the principled loss function used to train the RDM. This is in addition to the multiple RDMs and IBM which have no equivalents in FT. This sets our work apart from FT.
> >
> > - We showed superior performance against newer works like MLKD, CTKD, and LSKD which the reviewer requested (Table 14).
> >
> > Additionally, based on other reviewers' questions
> >
> > - We showed that proposed CIFD scales well with increasing student-teacher gap. In Table 13, we showed that distilling from a ResNet152 to ResNet18 model, CIFD provided a 0.36\% improvement over top-1 ImageNet accuracy over the nearest competitor DistKD. We also showed that CIFD monotonically improved the student accuracy with teacher size, something not observed in DistKD.
> >
> > - We also compared with other works like Masked Generative Distillation and Diffusion KD, and showed superior performance against both of them.
> >
> > - We showed that CIFD works well for CLIP like models where the teacher-student gap are larger than ResNets (Table 16 and 17). By outperforming CLIP specific distillation methods, we also showed the generality of the proposed idea.
> >
> >
> > We are very grateful for the time and effort from the reviewer to provide us with a review that has helped improve our paper. We are happy to engage further if they have any questions. Thank you again.

---

### Official Review · Reviewer_sfSd · 2024-07-16

**Soundness:** 2
**Presentation:** 2
**Contribution:** 2
**Rating:** 5
**Confidence:** 4

**Summary:**

The paper presents a new distillation method, CIFD, designed based on *Shannon’s Rate-Distortion theory* and  **Information Bottleneck Principle (IBP)*. CIFD contains Rate-Distortion Modules (RDM) for the teacher to substitute heavy Teacher Assistant (TA) and Information BottelNeck Module (IBM) for the student to mimic the features from several RDMs. Experiments demonstrate the effectiveness of the method.

**Strengths:**

1. The paper is organized well.
2. The experiments on CLIPs are good, verifying the broader effectiveness of the method.

**Weaknesses:**

My main concerns are from three aspects: **i) the story of the paper; ii) the reason why CIFD works; iii) insufficient experiments and comparisons.** Some concerns are mixed among the three aspects. And I will list them one by one.

1. **Insufficient experiments on verifying the basic settings of the paper.**
The story starts with *"When the teacher model is significantly larger than the student, previous works that utilize TAs induce high training costs."* I would believe the basic settings of this work as the teacher-student network pairs are in large parameter scale differences. From this point of view, the paper should contain more systematic experiments to verify the efficacy under this setting. Specifically, CIFD should be compared with previous methods on ImageNet with teacher-student network pairs with large different parameter scales, not just traditional ResNet-34 -> ResNet-18 and ResNet-50 -> MobileNet-V1.

2. **The trade-off between the story and the empirical solutions.** In my opinion, the paper is a little bit overdecorated and overclaimed. The author proposes many concepts, such as *Shannon’s Rate-Distortion theory* and  *Information Bottleneck Principle (IBP)*, and claims **"This is the first application of Shannon’s Rate-Distortion theory to aid knowledge distillation"**. I don't mean that the aforementioned statement is misleading.  But, if we go deeper into the design, the reason why the method works may come from the **noise-adding and noise-removing process**. Many previous works have verified that the above process could benefit the learning process in computer vision, like MIM and diffusion models, which have been empirical solutions.   In KD, there also exist distillation methods following MIM and diffusion models, like MGD and DiffKD. From this point of view, the authors should not claim ***"the first"*** only, but make a deeper analysis of related methods and make detailed comparisons. ***I strongly encourage the authors to make a good balance between the story and the verified empirical solutions.*** Even though it seems not as novel as this version, it would provide the readers with more useful knowledge and insights.

3. **The design may alter the network architecture of the student.** It seems that the IBM module would also be included in the validation stage. If my judgment is true, the added module (though lightweight) would also benefit the performance. Under such circumstances, the comparisons with previous methods, especially for light-weight models, are unfair.

**Questions:**

See Weaknesses.

**Limitations:**

Limitations are included in the main paper and the Appendix

---

> ### Author Rebuttal · Authors · 2024-08-07
>
> We thank the reviewer for their time and effort in providing feedback. Detailed response after summary.
>
> **Summary**
> - We directly addressed the concern on the efficacy of CIFD over large-student teacher gap. We distilled RN18 using RN152, RN101, RN50 (Table 13) as teachers and showed that proposed CIFD shows increasing student performance with teacher size. We also analyzed CLIP models (Table 16, 17) and found that it provides more improvement over baseline when the teacher student capacity gap is large.
> - We extended our discussion in Appendix C.1 to showcase that the Masked-Image-Modeling (MIM) objective is an approximation of an upper bound on the IB objective, whereas ours is a direct approximation of the IB objective. We compared and showed superior performance compared to Masked Generative Distillation (MGD).
> - We compared with DiffKD and showed our performance is superior. Further, it takes two methods, DistKD and DiffKD to achieve similar performance.
> - We clarify that we did not modify the student architecture to ensure fairness
>
> **Details**
>
> > Large teacher distillation experiments
>
> We trained a distillation from RN152, RN101, RN50 teachers to RN18 distillation using CIFD. DistKD [5] showed that RN152, RN101 to RN18 face the issue of large capacity gap (Table 3 of DistKD). The parameter ratio from largest teacher to student is 5.12. Experimental results are in Table 13. We observed that student performance increased consistently with teacher size and also outperformed the performance of DistKD. This directly addresses the reviewer's concern about large teacher student capacity gap.
>
> Additionally, we also compare CLIP models. Due to resource constraints, we compared the improvement provided by proposed CIFD over the nearest baseline CLIPKD [31]. Specifically, we compared the difference when a ViT-L-14 teacher was used to train ViT-B-16 and ViT-S-16 students. The parameter ratios are 2.8 and 6.9, respectively. Results are in Table 16 and 17. CIFD shows greater improvement over baseline when the teacher student ratio is large for 3 out of the 5 zero-shot classification datasets and almost always for the two zero-shot retrieval datasets. The capacity gap in CLIP-like models is much larger than the capacity gap in traditional KD settings such as RN34-18 and RN50-MNV1.
>
> > Connections of IBP to MIM and DiffKD
>
> We called the RDM modules as such because they capture different levels of information based on the rate constraint. Given existing work in learning based data compression [9, 34, 35], which explicitly work on Rate-Distortion objective (like we do), we felt appropriate to call this the RDM.
>
> We were aware of the similarity between MIM and IBP, hence we included appendix C.1 in our submission to address this. Building upon our insights in appendix C.1, we show that MIM and MGD objectives are upper bounds on the IBP objective. Consider MIM shown in Fig. 7a. The objective from the information bottleneck principle is $\min -I(X;\hat{U}) + \lambda I(T;\hat{U})$. The first term is usually approximated as a reconstruction error between $X$ and $\hat{U}$. Looking at the second term, we can write, $I(T;\hat{U}) \leq H(T)$, where $H$ signifies the entropy. Since $H(T)$ is not dependent on the neural network parameters, it can be dropped. This is the MIM objective, $\min -I(X;\hat{U})$ or minimize the reconstruction error. Similar simplification also holds for MGD. So both MIM and MGD minimize an upper-bound on the IBP got by dropping the second term. In our case, we compute an approximation for the second term, thus making it tighter to the original IBP objective. IBP works better because it forces the student to focus on those features necessary to predict the teacher embedding (first term) and remove information not useful in the prediction (second term), whereas in MIM and MGD the removal of information is not present. There have recently been works studying how IBP and generalization errors are connected ("How does information bottleneck help deep learning?," ICML 23) which provides support to the proposed idea. Further, our method outperforms the performance gains obtained from MGD [Yang et. al., 22] as seen in Table 14
>
> We have cited the work of DiffKD in related work. Based on our understanding (Fig. 3 in DiffKD), it appears extra modules are added during inference which changes the student architecture, which is not fair. However for completeness, we compare against it in Table 14. Proposed CIFD outperforms DiffKD except for top-1 accuracy in RN50 to MNV1 distillation. It takes the combination of two works DiffKD [25] and DistKD [5] to obtain better top-1 performance than CIFD in both RN34 to RN18 and RN50 to MNV1. However, our top-5 accuracy is better for both cases, indicating the competitiveness of proposed CIFD.
>
> We respectfully disagree with the characterization that CIFD is just noise addition and removal process and thus similar to MGD & DiffKD, latter only removes noise. **The key differentiator are the principled loss functions used to train the models in the presence of noise.** If we were to just add and remove noise from student's embeddings, then it would result in a simple, one-stage procedure, similar to MGD/DiffKD. The central theme of the paper is to control information flow during distillation and one of the best tools to accomplish it is by adding noise. Further, the **task accomplished by the noise in the teacher (mimic TAs) is different from the student model.** Finally, the effect of this is seen in the **performance gains obtained by our proposed method**
>
> > Student architecture changed?
>
> We do not modify the student architecture. We designate a layer in the unmodified student model as the bottleneck (usually the penultimate layer). The model's layers preceding the bottleneck acts as the IB encoder, and the layer(s) following as the decoder. Noise is added to the output of the bottleneck layer during training and noise addition is disabled during inference

---

> ### Comment · Reviewer_sfSd · 2024-08-12
> **Post Rebuttal comments by Reviewer sfSd**
>
> Thanks for your detailed response, which addresses my partial concerns.
>
> Here are my remaining concerns.
>
> I personally believe that lacking deep discussions about related techniques in the main paper is inappropriate. I encourage the author to add this content in the main paper, not just in the Appendix.
>
> Although the motivations seem novel, the main improvements are from multiple RDMs. Similar distillation designs have been explored in [1-3]. On the other hand, when reading the paper and the authors' response, I find no obvious technical flaws. Therefore, I'm a little bit confused about whether the effectiveness comes from the theoretical framework that the author claimed.
>
> As I think the above arguments are not strong enough to reject the paper, I would consider my final rating according to both the response and other reviewers' comments.
>
>
>
>
>
>
> [1] Chen, Yudong, et al. "Improved feature distillation via projector ensemble." Advances in Neural Information Processing Systems 35 (2022): 12084-12095.
>
> [2] Zhu, Xiatian, and Shaogang Gong. "Knowledge distillation by on-the-fly native ensemble." Advances in neural information processing systems 31 (2018).
>
> [3] Liu, Xiaolong, et al. "Norm: Knowledge distillation via n-to-one representation matching." arXiv preprint arXiv:2305.13803 (2023).

---

> ### Author Response · Authors · 2024-08-12
> **Response to Reviewer sfSd's post rebuttal comments**
>
> We again thank the reviewer again for their time and response. We greatly appreciate their interest in helping us improve the paper.
>
> > Moving discussion between IBP and MIM to main paper
> - We absolutely agree that having this discussion in the main paper will enrich it. In fact, it was in the main body of the initial draft. However, the space requirements proved very restrictive and we were forced to move the discussion to the appendix. We agree with you that it should be put back into the main paper and plan to do so in the final version.
>
> > Although the motivations seem novel, the main improvements are from multiple RDMs.
>
> **Difference between [1], [3] and our work.**
>
> While the multiple projection approaches has existed as pointed out in [1] (which is also cited as [20] in our paper), we performed comparisons specifically with [1] in the rebuttal pdf (Table 15). Note that *[1] and [3] use multiple projectors at the student* (see Fig. 1(c) in [1] and Fig. 1, right subfigure in [3]) and *our method uses multiple RDMs at the teacher*. Thus, these methods are quite different. For completeness we include comparisons with these methods in the table below. However, our method with 1 RDM and IBM performs close to [3] and the with 3 RDMs outperforms it. Additionally, our method with 3 RDMs and IBM outperforms [3] with 8 projectors at the student.
>
> To further answer your concern about whether the claimed method of RDM training is giving the performance improvement, we compare with Factor Transfer Kim et. al [28] in Table 15 (rebuttal PDF). Our proposed method with one RDM and no IBM outperforms FT [28], where the difference is the loss function used to train the RDM. This shows that the improvement is coming from the new principled loss function used to train the RDM network.
>
> **Difference between [2] and ours.**
>
> We have to note that [2] is for Online Distillation *without a teacher*. This setting itself is different from ours where use the presence of a teacher. While [2] does use an ensemble of projection modules at the teacher, the difference in setting used to train makes it hard to compare them. Specifically they use a gated ensemble of branch modules to simulate a teacher using the student's backbone. *In the below table we include [2] for reference only, the setting is different and they should not be compared directly.*
>
> ```
> +------------------------------+-------+-------+
> |     RN34 to RN18 on IN1k     | Top-1 | Top-5 |
> +------------------------------+-------+-------+
> |       FT (Kim et. al.)       | 71.43 | 90.29 |
> |           IFD [1]            | 71.94 | 90.68 |
> |  ONE [2] (No RN34 teacher)   | 70.55 | 89.59 |
> |           NORM [3]           | 72.14 |  ---  |
> |   ------------------------   | ----- | ----- |
> | Proposed (1 RDM) and no IBM  | 71.83 | 90.69 |
> | Proposed (1 RDM) and wt IBM  | 72.05 | 90.70 |
> | Proposed (3 RDMs) and wt IBM | 72.32 | 90.88 |
> +------------------------------+-------+-------+
> ```
>
> **Summary**
>
> We have shown over multiple experiments the efficacy of our proposed method. See Table 2 on CIFAR-100, Table 3 and Table 14 on ImageNet, Table 13 for large student-teacher gap in ImageNet, Table 5 on CLIP models. Specifically we have addressed the question if our proposed method is reason for performance improvement by disabling various parts of the proposed idea and show improved performance with existing methods (see Table 15). **These specifically showcase the importance of the proposed loss functions and their improved performance.**
>
> We again thank the reviewer for their time and effort. If you have any further questions, please feel free to reach out to us. Thank you!

---

> > ### Comment · Reviewer_sfSd · 2024-08-13
> > **Comments by Reviewer sfSd**
> >
> > Thank you very much for your response.
> >
> > As I am on the fence about this submission, I will make my final decision after discussing with other reviewers.

---

> > > ### Author Response · Authors · 2024-08-13
> > > **Thank you!**
> > >
> > > Thank you again for your time and effort in helping improve our paper.
> > >
> > > We wanted to specially acknowledge the question on large student-teacher capacity gap experiments that pushed us to provide stronger results during the rebuttal. It helped us show that proposed CIFD not only shows consistent improvement with increasing teacher size but also shows significant improvement over existing methods (+0.36% on ImageNet top-1 accuracy for ResNet152 to ResNet18 over existing DistKD), in Table 13. Further our methods showed similar trends for larger student-teacher gaps in CLIP like models (Tables 16 and 17).
> > >
> > > Additionally, based on your questions,
> > > - We showed that Masked-Image-Modeling objective is an upper-bound on the Information Bottleneck objective
> > > - We showed improved performance over works like MGD, DiffKD, IFD, and NORM.
> > > - We showed how our multiple RDM method which acts on the teacher embeddings is different from multi projector methods like [1] and [3] which act on student embeddings. Specifically, our RDMs are used to mimic Teacher Assistants, whereas projectors are used to better help the student model learn the teacher embedding by providing gradient via an ensemble of projectors. Additionally, we showed improved performance over  these methods.
> > >
> > > If you have any further questions, please feel free to ask and we are happy to discuss. If not, thank you again for your time and effort.

---

### Author Rebuttal · Authors · 2024-08-07

We thank all the reviewers for their time and effort for their feedback. We summarize the main points raised and our response. Detailed responses can be found in the reviewers' individual responses. Tables 13 - 17 and Fig. 7 are in the response PDF.

---

### Summary

 - Teacher Assistants have had significant success in Knowledge distillation in facilitating knowledge transfer between the larger teacher and the smaller teacher. However, they are expensive to train. To alleviate this we propose module called Rate-Distortion-Module (RDMs) to mimic teacher assistants by reusing the teacher embeddings. Since RDMs are only two to three layers, they are significantly less costly to train.

 - We propose the use of the Information Bottleneck Module (IBM) in the student model during training. We find that IBM on its own provides benefits but is also a crucial regularizer as the number of RDMs increases.

 - Across multiple large scale datasets and models like classification on ImageNet and CLIP, we show that our proposed method outperforms existing methods.

---

 ### Strengths as per the reviewers

- Multiple reviewers (sFSd and iLhT) appreciated the broad effectiveness of the proposed work. Specifically, they highlighted the contribution of the proposed idea in distilling CLIP like models.

- Reviewer vQtp appreciated the idea of using RDMs which in turn enable the proposed method to create cheaper Teacher Assistants by reusing the shallow modules of the teacher network.

---

### Major questions from the reviewers
While we respond to the individual queries in the reviewer specific rebuttals, here we list some of the important ones and our response.

- Reviewers sFSd and vQTp requested experimental results with large teacher student capacity gaps. To address this we conducted more experiments and our results show that CIFD works well when teacher student capacity gap is large. Details of the conducted experimented are as follows:
    1. We trained a ResNet18 model with ResNet34, ResNet50, ResNet101, and ResNet152 teachers (Table 13). We showed that CIFD trained student model showed consistent improvement with increasing teacher size (and hence increased capacity gaps) and comfortably outperformed DistKD [5] which experimented on the same combinations. These models were chosen because DistKD [5] showed that knowledge distillation in the ResNet101-ResNet18 and ResNet152-ResNet18 teacher-student combos suffer from the large capacity gap.
    2. Additionally, we also analyzed results with larger capacity gaps in the CLIP scenario. Here, CIFD based models showed more improvement over baseline (CLIPKD [31]) when the student teacher capacity gap was larger. Specifically, CIFD gives greater improvement when the capacity gap is larger for 3 out of 5 zero-shot classification datasets (Table 16) and almost always for two zero-shot retrieval datasets (Table 17) compared to the case when the capacity gap is smaller. Here, the parameter ratio between teacher to student is 6.9.

- Reviewer SfSd requested a comparison between Information Bottleneck Module and Masked-Image-Modeling (MIM) based distillation. Building up on our discussion in Appendix C.1 we provide novel insight that MIM minimization objective is an upper-bound on the Information Bottleneck objective (and hence, the IB objective is better). Further, we showed significant improvement over existing MIM based distillation, Masked Generative Distillation.

- Reviewer sFSd and iLhT requested comparisons with more recent works like MLKD, CTKD, LSKD, and Diffusion KD. In Table 13, we showcase that we outperform them all. In fact, it takes the combination of two works DiffKD [25] and DistKD [5] together to obtain results similar to ours.

---

### Decision · Program_Chairs · 2024-09-25

**Decision:**

Accept (poster)

**Comment:**

This submission leverages rate-distortion theory to formulate a new method which can replace computationally intensive teacher assistants and improve knowledge distillation process. The proposed method CIFD consists of two modules to be trained separately, namely rate-distortion module (RDM) conditioned on the penultimate-layer features from the teacher backbone and generative feature reconstruction with noise, and information bottleneck module (IBM) to guide the regularization of the student model given multiple RDMs. The submission was initially scored (4,5,4) by three knowledgeable reviewers, who recognized some positive aspects of the paper, **a)** exploring a new technical perspective on how to construct efficient and effective teacher assistant alternatives; **b)** the proposed method shows competitive performance; **c)** CLIP based settings are explored in experiments. Meanwhile, the reviewers also raised some major concerns about **1)** unconvincing motivation connected to teacher assistants; **2)** insufficient experiments, the model scale gap of teacher and student is not large (thus cannot well support its claimed motivation), and lacking sufficient comparison with recent knowledge distillation methods; **3)** somewhat unfair/inconsistent experimental settings; **4)** how performance gain is achieved, or say, the role of generative feature reconstruction with noise may play the key role instead of the proposed RDM; **5)** the computational complexity of RDMs in training process.

The authors provided detailed responses to these concerns, which are most recognized by reviewers. Finally, this submission has scores of (5,5,5) increased from initial (4,5,4), and all reviewers are on somewhat positive side. The AC read the paper, the reviews, the rebuttal and the reviewers' feedback, and agree with reviewers' assessment. To some degree, I also agree with the remained concern from reviewer sfSd, a more in-depth discussion of DRMs with existing generative knowledge distillation methods such as MGD and DiffKD as well as many-to-one knowledge distillation methods should be added into the paper, bettering positioning this work's contributions. Balancing above aspects of this submission, I recommend to accept (poster). The authors are encouraged to carefully consider the reviewers' comments/suggestions and their rebuttal in the final paper revision.